# Characterisation of the opposing effects of G6PD deficiency on cerebral malaria and severe malarial anaemia

Geraldine M Clarke[1,2*], Kirk Rockett[1,2,3*], Katja Kivinen[3], Christina Hubbart[1], Anna E Jeffreys[1], Kate Rowlands[1], Muminatou Jallow[4,5], David J Conway[4,6], Kalifa A Bojang[4], Margaret Pinder[4], Stanley Usen[4], Fatoumatta Sisay-Joof[4], Giorgio Sirugo[4], Ousmane Toure[7], Mahamadou A Thera[7], Salimata Konate[7], Sibiry Sissoko[7], Amadou Niangaly[7], Belco Poudiougou[7], Valentina D Mangano[8], Edith C Bougouma[9], Sodiomon B Sirima[9], David Modiano[8], Lucas N Amenga-Etego[10], Anita Ghansah[11], Kwadwo A Koram[11], Michael D Wilson[11], Anthony Enimil[12], Jennifer Evans[13,14], Olukemi K Amodu[15], Subulade Olaniyan[15], Tobias Apinjoh[16], Regina Mugri[17], Andre Ndi[17], Carolyne M Ndila[18], Sophie Uyoga[18], Alexander Macharia[18], Norbert Peshu[18], Thomas N Williams[18,19], Alphaxard Manjurano[20,21], Nuno Sepúlveda[21], Taane G Clark[6,21], Eleanor Riley[21], Chris Drakeley[20,21], Hugh Reyburn[20,21], Vysaul Nyirongo[22], David Kachala[23], Malcolm Molyneux[22,24], Sarah J Dunstan[25], Nguyen Hoan Phu[23,26], Nguyen Ngoc Quyen[23], Cao Quang Thai[23,26], Tran Tinh Hien[23,26,27], Laurens Manning[27], Moses Laman[27], Peter Siba[27], Harin Karunajeewa[28], Steve Allen[29], Angela Allen[30], Timothy ME Davis[28], Pascal Michon[27,31], Ivo Mueller[27,32,33], Síle F Molloy[1], Susana Campino[3], Angeliki Kerasidou[1,34], Victoria J Cornelius[1,2], Lee Hart[1], Shivang S Shah[1,35], Gavin Band[1,2], Chris CA Spencer[1], Tsiri Agbenyega[12,36], Eric Achidi[17], Ogobara K Doumbo[7], Jeremy Farrar[23,37], Kevin Marsh[18], Terrie Taylor[38], Dominic P Kwiatkowski[1,2,3*], MalariaGEN Consortium[1,3]

*For correspondence: gclarke@well.ox.ac.uk (GMC); krockett@well.ox.ac.uk (KRoc); dominic@well.ox.ac.uk (DPK)

[1]Wellcome Trust Centre for Human Genetics, University of Oxford, Oxford, United Kingdom; [2]MRC Centre for Genomics and Global Health, University of Oxford, Oxford, United Kingdom; [3]The Wellcome Trust Sanger Institute, Cambridge, United Kingdom; [4]Medical Research Council Unit The Gambia, Fajara, Gambia; [5]Edward Francis Small Teaching Hospital, Independence Drive, Banjul, Gambia; [6]Department of Pathogen Molecular Biology, London School of Hygiene and Tropical Medicine, London, United Kingdom; [7]Malaria Research and Training Centre, University of Bamako, Bamako, Mali; [8]University of Rome La Sapienza, Rome, Italy; [9]Centre National de Recherche et de Formation sur le Paludisme (CNRFP), Ouagadougou, Burkina Faso; [10]Navrongo Health Research Centre, Navrongo, Ghana; [11]Noguchi Memorial Institute for Medical Research, University of Ghana, Accra, Ghana; [12]Komfo Anoyke Teaching Hospital, Kumasi, Ghana; [13]Department of Molecular Medicine, Bernhard Nocht Institute for Tropical Medicine, Hamburg, Germany; [14]Kumasi Centre for Collaborative Research, Kumasi, Ghana; [15]University of Ibadan, Ibadan, Nigeria; [16]Department of Biochemistry and Molecular Biology, University of Buea, Buea, Cameroon; [17]Department of Medical Laboratory Sciences, University of Buea, Buea, Cameroon; [18]KEMRI-Wellcome Trust Research Programme, Kilifi, Kenya; [19]Department of Medicine, Faculty of Medicine, Imperial College, London, United Kingdom; [20]Joint Malaria Programme, Kilimanjaro Christian Medical College, Moshi, Tanzania; [21]Faculty of Infectious and Tropical Diseases, London School of Hygiene

and Tropical Medicine, London, United Kingdom; [22]Malawi-Liverpool Wellcome Trust Clinical Research Programme, University of Malawi, Blantyre, Malawi; [23]Oxford University Clinical Research Unit, University of Oxford, Ho Chi Minh City, Vietnam; [24]Liverpool School of Tropical Medicine, Pembroke Place, Liverpool, United Kingdom; [25]The Peter Doherty Institute for Infection and Immunity, The University of Melbourne, Melbourne, Australia; [26]Hospital for Tropical Diseases, Ho Chi Minh City, Viet Nam; [27]Papua New Guinea Institute of Medical Research, Goroka, Papua New Guinea; [28]University of Western Australia, Perth, Australia; [29]Swansea University, Swansea, United Kingdom; [30]Weatherall Institute of Molecular Medicine, Oxford University, Oxford, United Kingdom; [31]Faculty of Medicine and Health Sciences, Divine Word University, Madang, Papua New Guinea; [32]Walter and Eliza Hall Institute of Medical Research, Melbourne, Australia; [33]Barcelona Centre for International Health Research, Barcelona, Spain; [34]Nuffield Department of Population Health, The Ethox Centre, University of Oxford, Oxford, United Kingdom; [35]Laboratory of Malaria and Vector Research, National Institute of Allergy and Infectious Diseases, National Institutes of Health, Bethesda, United States; [36]Kwame Nkrumah University of Science and Technology, Kumasi, Ghana; [37]Nuffield Department of Clinical Medicine, Center for Tropical Medicine, Oxford University, Oxford, United Kingdom; [38]Blantyre Malaria Project at the College of Medicine, University of Malawi, Blantyre, Malawi

**Abstract** Glucose-6-phosphate dehydrogenase (G6PD) deficiency is believed to confer protection against *Plasmodium falciparum* malaria, but the precise nature of the protective effect has proved difficult to define as G6PD deficiency has multiple allelic variants with different effects in males and females, and it has heterogeneous effects on the clinical outcome of *P. falciparum* infection. Here we report an analysis of multiple allelic forms of G6PD deficiency in a large multi-centre case-control study of severe malaria, using the WHO classification of G6PD mutations to estimate each individual's level of enzyme activity from their genotype. Aggregated across all genotypes, we find that increasing levels of G6PD deficiency are associated with decreasing risk of cerebral malaria, but with increased risk of severe malarial anaemia. Models of balancing selection based on these findings indicate that an evolutionary trade-off between different clinical outcomes of *P. falciparum* infection could have been a major cause of the high levels of G6PD polymorphism seen in human populations.

## Introduction

Glucose-6-phosphate dehydrogenase (G6PD), encoded by the *G6PD* gene on chromosome X, is an enzyme that acts to control oxidative damage in red blood cells. G6PD deficiency is a common human genetic condition with many allelic variants causing different levels of enzyme deficiency, so that in some individuals it is asymptomatic whereas in others it can cause severe haemolytic anaemia (*Beutler, 1994*; *Beutler and Vulliamy, 2002*; *Beutler, 2008*; *Minucci et al., 2012*). The geographical distribution of G6PD deficiency corresponds to regions of the world where malaria is endemic, or has been in the relatively recent past, and it has been estimated that approximately 350 million people living in malaria-endemic countries are affected (*Allison, 1960*; *Motulsky, 1960*; *Howes et al., 2012*).

It was proposed over half a century ago, and is now widely accepted, that G6PD deficiency has risen to high frequency in malaria-endemic regions because it confers some level of resistance to malaria (*Allison, 1960*; *Luzzatto, 2015*). Evolutionary support for this hypothesis is provided by haplotype and sequence analyses indicating that *G6PD* variants have evolved independently, at a frequency that is inconsistent with random genetic drift, and over a period of time that is consistent

with the estimated time since the emergence of malaria (*Tishkoff et al., 2001*; *Sabeti et al., 2002*; *Saunders et al., 2002*; *Verrelli et al., 2002*; *Saunders et al., 2005*).

Although there is a large body of circumstantial evidence to suggest that G6PD deficiency confers resistance to malaria, there has been much debate about the precise nature of the protective effect. This is partly due to the genetic complexity of the locus, being X-linked with multiple deficiency alleles with the result that it has extensive allelic and phenotypic heterogeneity, and also to the fact that clinical and epidemiological studies have appeared to give conflicting results. An early study concluded that G6PD deficiency protects against *P. falciparum* malaria in heterozygous females (*Bienzle et al., 1972*) and this is supported by a number of more recent studies (*Manjurano et al., 2012*; *Sirugo et al., 2014*; *Luzzatto, 2015*; *Manjurano et al., 2015*; *Uyoga et al., 2015*). However other studies have indicated that the protective effect is present in both heterozygous females and hemizygous males (*Ruwende et al., 1995*; *Clark et al., 2009*; *Shah et al., 2016*) or that it is confined to hemizygous G6PD-deficient males (*Guindo et al., 2007*), or that there are no protective effects at all (*Johnson et al., 2009*; *Toure et al., 2012*).

The question of whether G6PD deficiency confers resistance to severe malaria, i.e. to life-threatening complications of *P. falciparum* infection, was recently examined in a multi-centre case-control study of much larger sample size than previous studies (*Malaria Genomic Epidemiology Network, 2014*). Standardised clinical definitions were used to categorise cases of severe malaria into sub-phenotypes, the main ones being cerebral malaria and severe malarial anaemia. It was found that both female heterozygotes and male hemizygotes for the main African form of G6PD deficiency (encoded by the derived allele of rs1050828 (*G6PD c.*202C>T)) had reduced risk of cerebral malaria, but this benefit was offset by increased risk of severe malarial anaemia in male hemizygotes and female homozygotes. By contrast, the sickle cell trait and blood group O were very strongly associated with reduced risk of both cerebral malaria and severe malarial anaemia in the same samples. (*Malaria Genomic Epidemiology Network, 2014*, *2015*). G6PD deficiency thus appears to have a more complex mode of action than other malaria-resistance factors, reducing the risk of some life-threatening complications of *P. falciparum* infection, but increasing the risk of others.

The present study had two main aims. The first aim was to perform a more comprehensive analysis of how an individual's level of G6PD deficiency affects the risk of severe malaria in general, and of cerebral malaria and severe malarial anaemia in particular. Previously published data are restricted to specific G6PD-deficient alleles, and do not consider the great allelic heterogeneity of this locus, or the fact that some allelic variants cause much more severe loss of G6PD enzyme function than others (*Beutler and Vulliamy, 2002*; *Minucci et al., 2012*). We therefore surveyed a broad range of G6PD-deficient alleles, using the World Health Organisation (WHO) system that classifies known allelic variants into different levels of severity according to the amount of enzyme function that is lost (*Yoshida et al., 1971*). The second aim of the study was to explore evolutionary and epidemiological models that are based on these new findings to re-examine the hypothesis that *P. falciparum* malaria is a major force for balancing selection of *G6PD*.

## Results

### Study population

Individuals with severe *P. falciparum* malaria and population controls were recruited in Burkina Faso, Cameroon, The Gambia, Ghana, Kenya, Malawi, Mali, Nigeria, Tanzania, Vietnam and Papua New Guinea as described elsewhere (*Malaria Genomic Epidemiology Network, 2008*, *2014*). After data curation and quality control, this analysis included 11,871 cases of severe malaria and 16,889 population controls. (*Table 1*, *Supplementary files 1A,B*). A total of 3,359 individuals had cerebral malaria only, 2,184 had severe malarial anaemia only, 741 had both cerebral malaria and severe malarial anaemia, and 5,587 cases fulfilled other WHO criteria for severe malaria (*World Health Organisation, 2000*). All of the severe malaria cases recruited in Africa were children: the median age of those with severe malarial anaemia was 1.8 years (interquartile range [IQR] 1.0–3.0) and that of those with cerebral malaria, 3.4 years (2.1–5.8). These differences in age, and variations in clinical patterns of disease at different geographical locations, were consistent with previous reports (*Slutsker et al., 1994*; *Snow et al., 1994*, *1997*; *Modiano et al., 1998*; *Newton and Krishna, 1998*;

**Table 1.** Clinical phenotype case counts, percentage of fatalities and descriptive statistics. Numbers are given for cases and controls of all severe malaria by study site, with the percentage of case fatalities shown in parentheses. Cases are further divided into those with cerebral malaria, severe malarial anaemia, both cerebral malaria and severe malarial anaemia, and other severe malaria. Percentage of males refers to cases and controls. CM, cerebral malaria; SMA, severe malarial anaemia.

| Study site | Case and control counts (% fatality) | | | | | | % males | Age in years (IQR) | | | |
|---|---|---|---|---|---|---|---|---|---|---|---|
| | CM | SMA | CM and SMA | Other severe malaria | Total cases | Total controls | | CM | SMA | All severe malaria | Controls* |
| Gambia | 815 (26) | 470 (5) | 139 (24) | 1,082 (7) | 2,506 (14) | 3,281 | 51.2 | 4.5 (3–7) | 2.3 (1.5–3.9) | 3.8 (2.2–5.9) | 0 |
| Mali | 78 (27) | 182 (8) | 69 (22) | 107 (12) | 436 (15) | 328 | 53.3 | 5 (3.5–7.2) | 2 (1.2–3.4) | 3 (1.7–5) | 3 (2–5) |
| Burkina Faso | 107 (20) | 38 (11) | 20 (20) | 681 (1) | 846 (5) | 721 | 54.0 | 4 (3–6) | 2.8 (2–4) | 3.8 (2–6) | 3 (2–4) |
| Ghana (Navrongo) | 21 (24) | 244 (2) | 14 (36) | 389 (3) | 668 (4) | 197 | 56.7 | 1.8 (1.1–2.4) | 1.1 (0.8–1.8) | 1.3 (0.9–2) | 1.2 (0.8–1.7) |
| Ghana (Kumasi) | 230 (10) | 548 (2) | 76 (13) | 635 (2) | 1,489 (4) | 2,027 | 52.2 | 2.5 (1.5–4) | 1.3 (0.8–2.5) | 2 (1–3.7) | 0 |
| Nigeria | 6 (17) | 7 (14) | 0 (0) | 64 (2) | 77 (4) | 40 | 57.0 | 1.7 (1–5.6) | 2.2 (1.6–4.5) | 3 (1.6–4) | 2.6 (1.1–3.9) |
| Cameroon | 39 (18) | 82 (7) | 8 (50) | 493 (2) | 622 (5) | 576 | 60.7 | 3 (1.4–4.2) | 2 (1.2–3.5) | 2.1 (1.2–4) | 21 (7.5–28) |
| Kenya | 901 (13) | 158 (8) | 213 (16) | 972 (8) | 2,244 (11) | 3,935 | 50.9 | 2.6 (1.6–3.8) | 1.8 (1–3) | 2.2 (1.2–3.6) | 0.5 (0.4–0.7) |
| Tanzania | 34 (41) | 179 (4) | 28 (29) | 186 (11) | 427 (11) | 455 | 49.0 | 2.3 (1.7–3.8) | 1.5 (0.9–2.5) | 1.7 (1.1–2.7) | 2.9 (2.1–3.9) |
| Malawi | 875 (15) | 129 (7) | 157 (20) | 219 (27) | 1380 (17) | 2,582 | 52.0 | 3 (2–4.7) | 2.2 (1.1–3.1) | 2.8 (1.8–4.2) | 0 |
| Vietnam | 209 (18) | 31 (6) | 8 (12) | 545 (9) | 793 (11) | 2,505 | 58.6 | 30 (22–42) | 24 (17.5–37.5) | 29 (22–41) | 0 |
| Papua New Guinea | 44 (5) | 116 (2) | 9 (11) | 214 (1) | 383 (2) | 242 | 55.3 | 3.8 (2.6–5) | 2.2 (1.3–3.2) | 2.9 (2.1–4.3) | 3.3 (2.2–4.8) |
| Total | 3,359 (18) | 2,184 (5) | 741 (20) | 5,587 (6) | 11,871 (10) | 16,889 | 53.1 | 3.4 (2.1–5.8) | 1.8 (1–3) | 2.8 (1.5–5) | 0 (0–0.7) |

*Four sites used cord blood controls (i.e. age 0).

*Marsh and Snow, 1999*; *Pasvol, 2005*; *Reyburn et al., 2005*; *Idro et al., 2006*; *Okiro et al., 2009*; *Thuma et al., 2011*). The overall case fatality rate was 10%.

## Survey of G6PD variation in the study population

We identified a set of 70 single nucleotide polymorphisms (SNPs) of functional or genetic relevance in a 560 kb region spanning the *G6PD* locus and part of the overlapping *IKBKG* gene (encoding Inhibitor of Kappa Light Polypeptide Gene Enhancer in B-Cells) (see Materials and methods). Of these, five SNPs failed quality control and six were monomorphic (*Supplementary file 1C*). The remaining 59 SNPs showed variation in allele frequency between study sites, with a median $F_{ST}$ of 0.024 (range 0 to 0.67). Fifteen SNPs expressed phenotypically deficient G6PD activity according to the WHO classification system (*Beutler and Vulliamy, 2002*; *Minucci et al., 2012*) , in which class I represents the most severe deficiency mutation and class IV the least (*Table 2*). The most common deficiency haplotype in sub-Saharan Africa, often referred to as G6PD A–, is composed of derived alleles at class III variant rs1050828 (G6PD c.202C>T, here referred to as G6PD+202) and class IV variant rs1050829 (G6PD c.376T>C, here referred to as G6PD+376). These two alleles, here referred to as G6PD+202T and G6PD+376C, were observed at all of the African study sites in strong linkage disequilibrium, with average allele frequencies of 15% and 39%, respectively, but neither was present in Papua New Guinea or Vietnam. Four other deficiency alleles were observed at >1% frequency in specific populations: the class III variant rs76723693 (G6PD c.968T>C) and the class II variant rs5030872 (G6PD c.542A>T) had deficiency allele frequencies of 7% and 1.5%, respectively, in The Gambia; and the class II variants rs137852327 and rs78365220 had deficiency allele frequencies of 1.4% and 3.7%, respectively, in Papua New Guinea. All of other G6PD deficiency alleles observed in this study were at <1% frequency in all populations. The only class I deficiency allele was at CM973154, where it was present in just three Gambian individuals.

## Analysis of association with individual variants

As reported previously (*Malaria Genomic Epidemiology Network, 2014*), G6PD+202T was associated with an increased risk of severe malarial anaemia (odds ratio [OR] = 1.18, p = $7.5 \times 10^{-5}$) and a

decreased risk of cerebral malaria (OR = 0.91, p = $7.2 \times 10^{-3}$) in males and females combined under an additive genetic model (*Table 3* and *Supplementary files 1D–F*). Increased risk of severe malarial anaemia was observed in male hemizygotes (OR = 1.48, p = $6.6 \times 10^{-5}$) and female homozygotes (OR = 1.86, p = $3.4 \times 10^{-3}$); and there was a trend for decreased risk of cerebral malaria in male hemizygotes (OR = 0.82, p = 0.02) and female heterozygotes (OR = 0.87, p = 0.05).

To investigate other SNPs, we began by testing individual variants for association with severe malaria, cerebral malaria and severe malarial anaemia across all study populations. This analysis assumed fixed effects across populations, under a variety of modes of inheritance, for males and females separately (see Materials and methods). Six variants other than G6PD+202 showed evidence of association with severe malarial anaemia (p<$1 \times 10^{-3}$) but they were all in linkage disequilibrium with G6PD+202 and these associations disappeared after adjusting for the known effects of G6PD +202 (*Figure 1*, *Table 4*). In these analyses, we corrected for the effects of the sickle-cell trait because of its strong protective effect against both cerebral malaria and severe malarial anaemia. We also examined the HbC variant, and the *ABO*, *ATP2B4,* and *FREM3/GYPE* loci, which have well-validated protective effects against severe malaria (*Malaria Genomic Epidemiology Network, 2014*, *2015*) and found no evidence that they affected the association of G6PD variants with severe malaria (*Figure 1—figure supplement 3*). The available data did not allow us to analyse $\alpha$ thalassemia across all study sites, but recent analyses at individual study sites have shown no evidence of interaction between the malaria-protective effects of G6PD deficiency and $\alpha$ thalassemia (*Manjurano et al., 2015*; *Uyoga et al., 2015*).

The above analyses assume that variants have the same effects in different locations, but variation in patterns of disease association can arise due to a range of genetic, biological and environmental factors. We therefore evaluated different models of genetic association allowing for heterogeneity of effect at different locations and on different sub-phenotypes (Materials and methods; *Figure 1—figure supplement 1*). As expected G6PD+202T showed strong evidence of heterogeneity of effect on different sub-phenotypes, with more than 90% of the posterior weight resting on models indicating different effects on cerebral malaria and severe malarial anaemia. Evidence for heterogeneity of effect was also seen at rs73573478, which showed variation of effect at different locations rather

**Table 2.** WHO-classified G6PD deficiency alleles observed in this study. Locus refers to GRCh37, dbSNP137 and Ensembl build 84. WHO refers to grade of G6PD deficiency based on the WHO classification scheme. Allele frequency is calculated in population controls. GM Gambia; ML, Mali; BF, Burkina Faso; GH-N, Ghana (Noguchi); GH-K, Ghana (Kumasi); NG Nigeria; CM, Cameroon; KY, Kenya; TZ, Tanzania; MW, Malawi; VN, Vietnam; PNG, Papua New Guinea.

| SNP | Locus | Base change | WHO | Allele frequency | | | | | | | | | | | |
| | | | | GM | ML | BF | GH-N | GH-K | NG | CM | KY | TZ | MW | VN | PNG |
| --- | --- | --- | --- | --- | --- | --- | --- | --- | --- | --- | --- | --- | --- | --- | --- |
| CM973154 | 153760261 | A/C | I | 0.0004 | 0.0000 | 0.0000 | 0.0000 | 0.0000 | 0.0000 | 0.0000 | 0.0000 | 0.0000 | 0.0000 | 0.0000 | 0.0000 |
| rs72554665 | 153760484 | C/A | II | 0.0004 | 0.0000 | 0.0000 | 0.0000 | 0.0000 | 0.0000 | 0.0000 | 0.0000 | 0.0000 | 0.0000 | 0.0055 | 0.0000 |
| CM920290 | 153760605 | G/A | II | 0.0002 | 0.0000 | 0.0000 | 0.0000 | 0.0000 | 0.0000 | 0.0000 | 0.0000 | 0.0000 | 0.0000 | 0.0014 | 0.0028 |
| rs137852342 | 153761184 | G/A | III | 0.0000 | 0.0020 | 0.0000 | 0.0000 | 0.0000 | 0.0000 | 0.0000 | 0.0000 | 0.0000 | 0.0000 | 0.0014 | 0.0000 |
| rs76723693 | 153761240 | 968T>C | III | 0.0695 | 0.0082 | 0.0009 | 0.0000 | 0.0003 | 0.0000 | 0.0000 | 0.0000 | 0.0000 | 0.0000 | 0.0000 | 0.0000 |
| rs137852327 | 153761337 | C/T | II | 0.0006 | 0.0000 | 0.0000 | 0.0000 | 0.0000 | 0.0000 | 0.0000 | 0.0000 | 0.0000 | 0.0000 | 0.0096 | 0.0141 |
| CM014189 | 153761820 | T/A | II | 0.0004 | 0.0000 | 0.0000 | 0.0000 | 0.0008 | 0.0000 | 0.0000 | 0.0004 | 0.0000 | 0.0000 | 0.0000 | 0.0000 |
| rs137852328 | 153762340 | 680G>T | III | 0.0002 | 0.0000 | 0.0000 | 0.0000 | 0.0000 | 0.0000 | 0.0000 | 0.0000 | 0.0000 | 0.0000 | 0.0000 | 0.0000 |
| rs137852330 | 153762605 | G/A | II | 0.0004 | 0.0000 | 0.0000 | 0.0000 | 0.0000 | 0.0000 | 0.0000 | 0.0000 | 0.0000 | 0.0000 | 0.0003 | 0.0000 |
| rs5030868 | 153762634 | 563G>A | II | 0.0000 | 0.0000 | 0.0000 | 0.0000 | 0.0000 | 0.0000 | 0.0003 | 0.0000 | 0.0000 | 0.0000 | 0.0000 | 0.0085 |
| rs5030872 | 153762655 | 542A>T | II | 0.0145 | 0.0041 | 0.0009 | 0.0000 | 0.0003 | 0.0000 | 0.0000 | 0.0000 | 0.0000 | 0.0000 | 0.0000 | 0.0000 |
| CM970547 | 153763462 | G/A | II | 0.0008 | 0.0000 | 0.0000 | 0.0000 | 0.0000 | 0.0000 | 0.0000 | 0.0000 | 0.0000 | 0.0000 | 0.0000 | 0.0000 |
| rs78365220 | 153763485 | A/G | II | 0.0002 | 0.0000 | 0.0000 | 0.0000 | 0.0000 | 0.0000 | 0.0000 | 0.0000 | 0.0000 | 0.0000 | 0.0000 | 0.0366 |
| rs1050829 | 153763492 | 376T>C | IV | 0.3206 | 0.4132 | 0.4186 | 0.4476 | 0.4267 | 0.5574 | 0.3429 | 0.4060 | 0.3787 | 0.3920 | 0.0000 | 0.0000 |
| rs1050828 | 153764217 | 202C>T | III | 0.0273 | 0.1616 | 0.1447 | 0.1713 | 0.1847 | 0.3167 | 0.1075 | 0.1930 | 0.2071 | 0.1988 | 0.0000 | 0.0000 |

than on different sub-phenotypes. For reasons of sample size, we did not conduct a detailed analysis of sub-types of severe malaria other than cerebral malaria and severe malaria, but summary statistics are available in *Figure 1—source data 1* and *2*.

In this dataset, the median age of those with severe malarial anaemia was 1.8 years and that of those with cerebral malaria was 3.4 years (*Table 1*). This is consistent with previous studies of severe malaria in African children, which have established that severe malarial anaemia tends to occur at a younger age than cerebral malaria (*Newton and Krishna, 1998*; *Taylor et al., 2006*). We therefore investigated the possible effect of age on associations with G6PD deficiency, dividing cases of severe malaria into those occurring within two main groups of age <2 years or 2–6 years. In both age groups, assuming an additive genetic model, the G6PD+202T allele was associated with reduced risk of cerebral malaria (<2 years, OR = 0.93, 95% CI = 0.82–1.05; 2–6 years, OR = 0.88, 95% CI = 0.81–0.96) and increased risk of severe malarial anaemia (<2 years, OR = 1.16, 95% CI = 1.05–1.28; 2–6 years, OR = 1.20, 95% CI = 1.06–1.36). We conclude from these findings, and from

**Table 3.** G6PD+202 best model association test results. p values and odds ratios, with 95% confidence interval, for association at G6PD+202 with all cases of severe malaria and with the two sub-types cerebral malaria and severe malarial anaemia, for all samples combined and separately for males and females. Results are shown for all sites combined and are adjusted for sickle-cell trait, ethnicity and sex.

| Sample | Case phenotype | Case frequency* | Control frequency* | Model[†] | Model OR (95% CI) | P | Sites excluded[‡] |
|---|---|---|---|---|---|---|---|
| All | Cerebral malaria | 0.13 (2,528/337/231) | 0.15 (11,155/1,701/1,229) | A | 0.91 (0.85–0.97) | $7.22 \times 10^{-3}$ | VN, PNG |
| | | 0.13 (2,528/337/231) | 0.15 (11,155/1,701/1,229) | D | 0.86 (0.77–0.95) | $4.54 \times 10^{-3}$ | VN, PNG |
| | | 0.13 (2,528/337/231) | 0.15 (11,155/1,701/1,229) | R | 0.86 (0.74–1) | 0.06 | VN, PNG |
| | Severe malarial anaemia | 0.17 (1,566/212/256) | 0.15 (11,155/1,701/1,229) | A | 1.18 (1.09–1.28) | $7.48 \times 10^{-5}$ | VN, PNG |
| | | 0.17 (1,566/212/256) | 0.15 (11,155/1,701/1,229) | D | 1.21 (1.06–1.38) | $5.44 \times 10^{-3}$ | VN, PNG |
| | | 0.17 (1,566/212/256) | 0.15 (11,155/1,701/1,229) | R | 1.53 (1.29–1.82) | $1.19 \times 10^{-6}$ | VN, PNG |
| | Severe malaria | 0.14 (8,536/1,138/986) | 0.15 (11,155/1,701/1,229) | A | 1.01 (0.97–1.06) | 0.65 | VN, PNG |
| | | 0.14 (8,536/1,138/986) | 0.15 (11,155/1,701/1,229) | D | 0.98 (0.92–1.05) | 0.58 | VN, PNG |
| | | 0.14 (8,536/1,138/986) | 0.15 (11,155/1,701/1,229) | R | 1.09 (0.99–1.2) | 0.08 | VN, PNG |
| Females | Cerebral malaria | 0.14 (1,145/337/38) | 0.15 (4,933/1,701/168) | A | 0.91 (0.81–1.03) | 0.15 | VN, PNG |
| | | 0.14 (1,145/337/38) | 0.15 (4,933/1,701/168) | D | 0.88 (0.77–1.01) | 0.08 | VN, PNG |
| | | 0.14 (1,145/337/38) | 0.15 (4,933/1,701/168) | H | 0.87 (0.75–1) | 0.05 | VN, PNG |
| | | 0.14 (1,145/337/38) | 0.15 (4,933/1,701/168) | R | 1.06 (0.73–1.54) | 0.75 | VN, PNG |
| | Severe malarial anaemia | 0.16 (673/212/40) | 0.15 (4,933/1,701/168) | A | 1.11 (0.95–1.29) | 0.19 | VN, PNG |
| | | 0.16 (673/212/40) | 0.15 (4,933/1,701/168) | D | 1.04 (0.87–1.24) | 0.68 | VN, PNG |
| | | 0.16 (673/212/40) | 0.15 (4,933/1,701/168) | H | 0.93 (0.77–1.13) | 0.47 | VN, PNG |
| | | 0.16 (673/212/40) | 0.15 (4,933/1,701/168) | R | 1.86 (1.23–2.81) | $3.40 \times 10^{-3}$ | VN, PNG |
| | Severe malaria | 0.14 (3,729/1,138/132) | 0.15 (4,933/1,701/168) | A | 0.95 (0.88–1.03) | 0.21 | VN, PNG |
| | | 0.14 (3,729/1,138/132) | 0.15 (4,933/1,701/168) | D | 0.92 (0.84–1.01) | 0.07 | VN, PNG |
| | | 0.14 (3,729/1,138/132) | 0.15 (4,933/1,701/168) | H | 0.9 (0.82–0.99) | 0.03 | VN, PNG |
| | | 0.14 (3,729/1,138/132) | 0.15 (4,933/1,701/168) | R | 1.13 (0.88–1.45) | 0.33 | VN, PNG |
| Males | Cerebral malaria | 0.12 (1,379/–/193) | 0.15 (6,207/–/1,058) | M | 0.82 (0.69–0.98) | 0.03 | NG, VN, PNG |
| | Severe malarial anaemia | 0.19 (893/–/216) | 0.15 (6,222/–/1,061) | M | 1.48 (1.22–1.8) | $6.58 \times 10^{-5}$ | VN, PNG |
| | Severe malaria | 0.15 (4,807/–/854) | 0.15 (6,222/–/1,061) | M | 1.08 (0.97–1.2) | 0.14 | VN, PNG |

*Derived allele frequency and counts of wild-type homozygotes, heterozygotes and derived homozygotes, where male hemizygotes are counted as female homozygotes.

[†]Models are additive (A), dominant (D), recessive (R) heterozygous model (H) or male hemizygous (M).

[‡]Sites at which an SNP is monomorphic in either cases or controls are excluded from the combined analysis. NG, Nigeria; VN, Vietnam; PNG, Papua New Guinea.

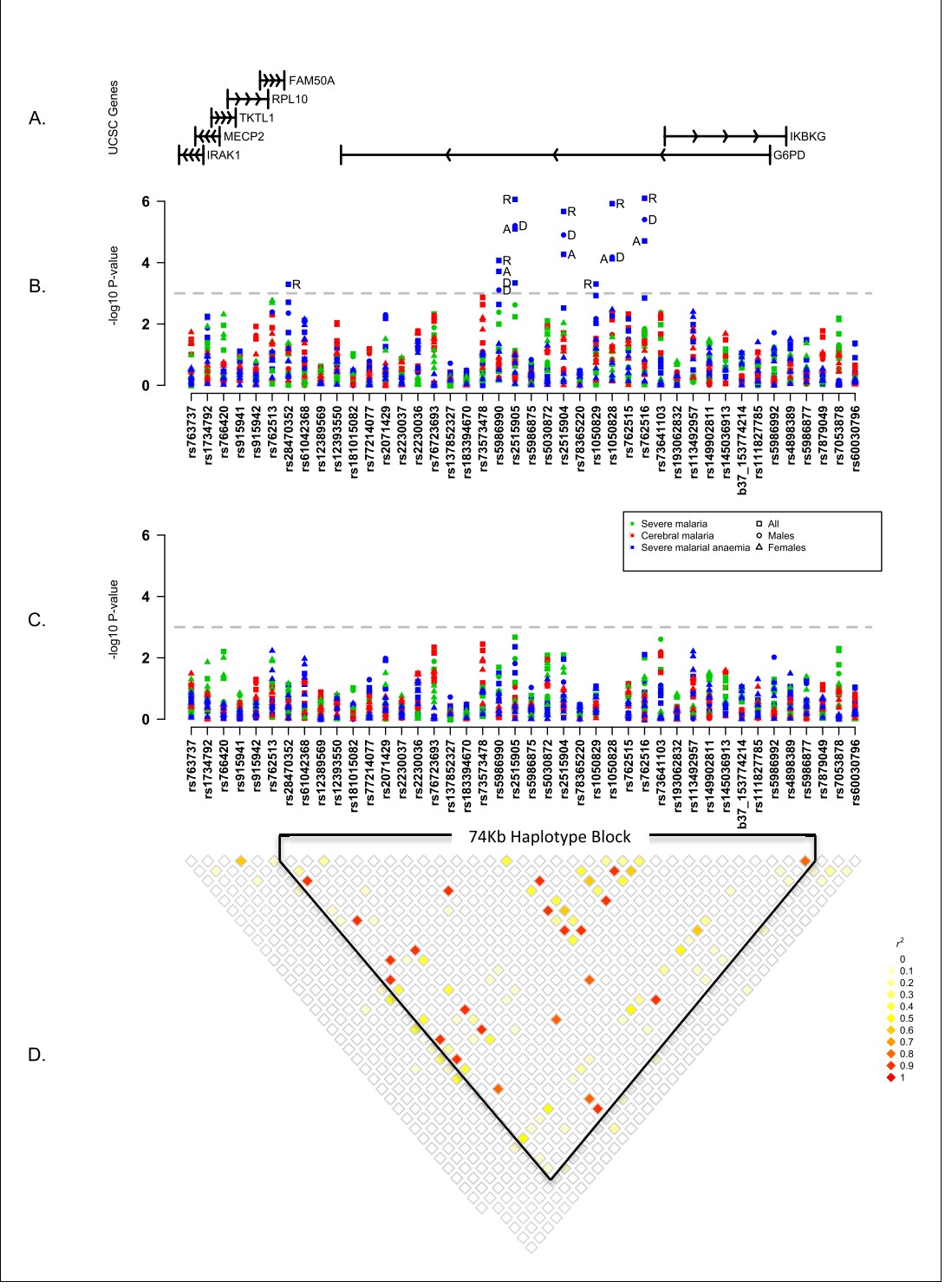

**Figure 1.** Summary of single SNP tests of association at all sites combined. (**A**) Schematic of genes across the genotyped region plotted relative to evenly spaced positions of SNPs in (B). (**B**) Manhattan plot showing the results of all single SNP tests of association with all severe malaria and with cerebral malaria and severe malarial anaemia for males, females and all individuals combined for models of association with additive, dominant, recessive and heterozygous (females only) modes of inheritance. Mode of inheritance is indicated as additive (A), dominant (D), recessive (R) or heterozygous (H) only for SNPs with a *P* value <1×10⁻³, as indicated by the horizontal dashed grey line. See *Figure 1—source data 1* for detailed association results. (**C**) As for (B), but results are adjusted for additive effects at G6PD+202. See *Figure 1—source data 2* for detailed association

*Figure 1 continued on next page*

*Figure 1 continued*

results with adjustment for G6PD+202. (**D**) Pairwise $r^2$ between pairs of SNPs in control individuals. Samples are excluded from analysis at sites where an SNP is monomorphic; this accounts for a variation in sample size across SNPs. Results for SNPs that are monomorphic or extremely rare across all sites (maximum minor allele frequency <0.01) are not shown. All results are adjusted for gender, ethnicity and the sickle-cell trait. See *Figure 1—figure supplement 1* for detailed forest plots of association at G6PD+202. See *Figure 1—figure supplement 2* for a summary of genetic heterogeneity of cerebral malaria and severe malarial anaemia within and across African sites.

The following source data and figure supplements are available for figure 1:

**Source data 1.** Single SNP association test results.

**Source data 2.** Single SNP association test results with adjustment for additive effect of G6PD+202.

**Figure supplement 1.** Genetic heterogeneity of severe malaria sub-phenotypes within and across African sites.

**Figure supplement 2.** Association between G6PD deficiency and severe malaria sub-phenotypes stratified by age.

**Figure supplement 3.** Association between G6PD deficiency and severe malaria with and without adjustment for additional genetic factors.

more detailed estimates of age-specific effects shown in *Figure 1—figure supplement 2*, that the effects of G6PD on the risk of cerebral malaria and severe malarial anaemia are not significantly affected by age.

## Analysis of association with G6PD-deficiency score

Most of the G6PD-deficiency variants identified in this study occur at low frequency or are restricted to specific populations, so we have relatively low statistical power to detect association with individual variants other than G6PD+202 and G6PD+376. One way of overcoming this problem is to amalgamate the evidence across multiple variants by considering their phenotypic effects. Although we do not have information on the precise phenotypic effects of each of the variants observed in this study, we can make an approximate estimate of an individual's level of G6PD deficiency based on their genotypes, using the WHO classification of functional severity of known mutations (*Table 2*). All of the cases and controls were assigned a G6PD deficiency score (G6PDd score), defined as the estimated level of G6PD deficiency for that individual, ranging from 0% (no loss of function) to 100% (total loss of function), and calculated using the scoring system shown in *Table 5* (see Materials and methods).

The mean G6PDd score was 13.5% in controls, 13% in cerebral malaria cases and 16.9% in severe malarial anaemia cases (*Supplementary file 1J*). If individuals carrying G6PD+202T are excluded, the mean G6PDd score was 6% in controls, 5.6% in cerebral malaria cases and 7.1% in severe malarial anaemia cases. This pattern of a higher mean G6PDd score in severe malarial anaemia cases compared to that in controls and cerebral malaria cases was observed in both males and females and at all sites other than Nigeria, Tanzania and Papua New Guinea (*Supplementary file 1J*). The most severe G6PDd score observed was 95%: this was found in 95 individuals, mainly from The Gambia, Vietnam and Papua New Guinea, who were homozygous for a class II mutation (*Supplementary files 1G–I*).

We divided the G6PDd score into five categories: normal (G6PDd score = 0); low (0 < G6PDd score < 25); medium (25 $\leq$ G6PDd score < 50); high (50 $\leq$ G6PDd score < 85); and very high (G6PDd score $\geq$ 85). (*Table 5* and Materials and methods.) In this system, the majority of G6PD +202T heterozygotes were categorized as medium and all G6PD+202T homozygotes, including male hemizygotes, were categorized as high. Only individuals carrying less severe WHO class IV mutations were categorized as low and only individuals homozygous for more severe WHO class II mutations had a very high G6PDd score. There were 16,236 individuals in the normal category, 6,633 in low, 3,311 in medium, 2,485 in high, and 95 with very high G6PDd score. No significant

**Table 4.** Association signals at loci previously reported to be in association with severe malaria and at loci with the strongest association signals in all individuals at all sites combined, after adjustment for G6PD+202. Odds Ratios (OR), 95% Confidence Intervals (95% CI) and P values (P) for the optimal genetic model for association with all severe malaria and with two sub-types, cerebral malaria and severe malarial anaemia, at loci previously reported to be associated with severe malaria or where the best model association P value is <0.01 in fixed effect tests of association at all sites combined. Results are only shown for loci with a minor allele frequency in controls of >1% or count of >5: this excludes previously reported loci rs137852328 and rs5030868. Signals are adjusted for the sickle-cell trait, ethnicity, sex and G6PD +202.

| SNP | Locus* | Base change | Control frequency† | Case phenotype | Case frequency† | Model‡ | Model OR(95% CI) | P | Sites excluded§ |
|---|---|---|---|---|---|---|---|---|---|
| **Previously reported loci** | | | | | | | | | |
| rs76723693 | 153761240 | 968T>C | 0.07 (3,254/210/129) | Cerebral malaria | 0.05 (835/36/20) | D | 0.65 (0.48–0.87) | $4.60\times10^{-3}$ | All except GM, ML |
| | | | 0.07 (2,930/208/127) | Severe malarial anaemia | 0.07 (421/29/20) | A | 1.06 (0.85–1.31) | 0.63 | All except GM |
| | | | 0.07 (3,254/210/129) | Severe malaria | 0.05 (2,737/128/70) | A | 0.83 (0.73–0.95) | $6.83\times10^{-3}$ | All except GM, ML |
| rs5030872 | 153762655 | 542A>T | 0.01 (3,530/48/24) | Cerebral malaria | 0.004 (885/5/1) | A | 0.42 (0.21–0.82) | $1.15\times10^{-2}$ | All except GM, ML |
| | | | 0.01 (3,530/48/24) | Severe malarial anaemia | 0.02 (636/7/6) | R | 1.62 (0.62–4.26) | 0.33 | All except GM, ML |
| | | | 0.01 (6,258/50/24) | Severe malaria | 0.01 (5,234/19/12) | D | 0.56 (0.36–0.85) | $6.99\times10^{-3}$ | All except GM, ML, BF |
| rs1050829 | 153763492 | 376T>C | 0.39 (7,039/3,244/3,771) | Cerebral malaria | 0.36 (1,620/710/746) | A | 0.97 (0.92–1.03) | 0.30 | VN, PNG |
| | | | 0.39 (7,039/3,244/3,771) | Severe malarial anaemia | 0.42 (955/427/629) | R | 1.12 (0.98–1.28) | 0.08 | VN, PNG |
| | | | 0.39 (7,039/3,244/3,771) | Severe malaria | 0.39 (5,332/2,278/2,966) | R | 1.05 (0.98–1.13) | 0.17 | VN, PNG |
| **Strongest association signals** | | | | | | | | | |
| rs766420 | 153554404 | C/G | 0.57 (5,333/3,529/7,792) | Cerebral malaria | 0.60 (975/715/1628) | D | 1.02 (0.93–1.11) | 0.74 | None |
| | | | 0.57 (5,333/3,529/7,792) | Severe malarial anaemia | 0.64 (550/443/1,168) | R | 0.97 (0.87–1.09) | 0.63 | None |
| | | | 0.57 (5,333/3,529/7,792) | Severe malaria | 0.6 (3,462/2,464/5,812) | R | 0.93 (0.88–0.98) | $6.20\times10^{-3}$ | None |
| rs73573478 | 153761564 | G/A | 0.10 (11,710/1,311/740) | Cerebral malaria | 0.12 (2,538/330/196) | D | 1.18 (1.06–1.32) | $3.30\times10^{-3}$ | NG, VN, PNG |
| | | | 0.10 (11,745/1,315/740) | Severe malarial anaemia | 0.10 (1,721/168/117) | D | 0.94 (0.8–1.09) | 0.40 | VN,PNG |
| | | | 0.10 (11,745/1,315/740) | Severe malaria | 0.11 (8,904/990/660) | R | 1.11 (0.99–1.25) | 0.08 | VN, PNG |
| rs2515905 | 153762075 | G/A | 0.19 (10,201/2,061/1,611) | Cerebral malaria | 0.18 (2,315/432/342) | R | 1.13 (0.94–1.36) | 0.20 | VN, PNG |
| | | | 0.19 (10,201/2,061/1,611) | Severe malarial anaemia | 0.21 (1,444/265/302) | R | 1.39 (1.11–1.75) | $4.36\times10^{-3}$ | VN, PNG |
| | | | 0.19 (10,201/2,061/1,611) | Severe malaria | 0.19 (7,861/1,423/1,325) | R | 1.21 (1.07–1.37) | $2.12\times10^{-3}$ | VN,PNG |
| rs73641103 | 153769889 | G/A | 0.01 (11,932/183/82) | Cerebral malaria | 0.02 (2,774/46/36) | R | 1.83 (1.21–2.77) | $4.20\times10^{-3}$ | ML, BF, NG, CM, VN, PNG |
| | | | 0.02 (13,485/206/92) | Severe malarial anaemia | 0.02 (1,961/28/17) | R | 1.56 (0.9–2.7) | 0.11 | NG, VN, PNG |
| | | | 0.02 (13,522/208/93) | Severe malaria | 0.02 (10,379/150/94) | R | 1.51 (1.11–2.05) | $8.11\times10^{-3}$ | VN, PNG |
| rs7053878 | 153834100 | T/A | 0.06 (15,249/739/597) | Cerebral malaria | 0.06 (2,996/153/134) | A | 1.09 (0.99–1.21) | 0.09 | GH-N, NG |
| | | | 0.06 (15,437/744/600) | Severe malarial anaemia | 0.09 (1,920/77/134) | D | 1.02 (0.82–1.28) | 0.84 | NG |
| | | | 0.06 (15,437/744/600) | Severe malaria | 0.07 (10,505/475/622) | A | 1.1 (1.03–1.17) | $6.98\times10^{-3}$ | NG |

*Locus refers to NCBI Build 37.

†Derived allele frequency and counts of wild-type homozygotes, heterozygotes and derived homozygotes where male hemizygotes are counted as female homozygotes.

‡Models are additive (A), dominant (D) or recessive (R).

§Sites at which an SNP is monomorphic in either cases or controls are excluded from the combined analysis. GM, Gambia; ML, Mali; BF, BurkinaFaso; GH_N, Ghana (Noguchi); NG, Nigeria; CM, Cameroon; VN, Vietnam; PNG, Papua New Guinea.

effects on cerebral malaria or severe malarial anaemia were observed at low G6PDd score. Individuals with medium G6PDd score had significant protection against cerebral malaria (OR = 0.86, p = 0.04). Individuals with high G6PDd score had a stronger level of protection against cerebral malaria (OR 0.8, p = 3.0x10$^{-3}$) and an increased risk of severe malarial anaemia (OR = 1.49, p = 2.2x10$^{-6}$). Individuals with very high G6PDd score showed the strongest level of protection against cerebral malaria (OR = 0.28, p = 0.03) and an increased risk of severe malarial anaemia (OR = 1.44, p = 0.36), although the latter was not significant due to the smaller sample numbers (*Table 6* and *Figure 2*).

We can simplify the above findings by assuming a linear relationship between the log odds of disease and the G6PDd score. In this statistical model, each 10% increase in the G6PDd score decreased the risk of cerebral malaria by approximately 4% (OR = 0.96, p = 5.6x10$^{-5}$) and increased the risk of severe malarial anaemia by approximately 5% (OR = 1.05, p = 1.4x10$^{-4}$) (*Supplementary file 1J*). The effect of G6PDd score on cerebral malaria appeared to be the same for males (OR = 0.96, p = 1.6x10$^{-3}$) and females (OR = 0.96, p = 1.4x10$^{-2}$), whereas its effect on severe malarial anaemia was more pronounced for males (OR = 1.05, p = 1.8x10$^{-4}$) than for females (OR = 1.03, p = 0.12).

## Models for balancing selection from opposing effects on cerebral malaria and severe malarial anaemia

These findings provide an opportunity to re-evaluate the evolutionary factors that cause G6PD deficiency to be maintained in human populations. Several lines of evidence indicate that the G6PD +202T allele has been under recent positive selection (*Tishkoff et al., 2001*; *Sabeti et al., 2002*; *Tripathy and Reddy, 2007*) and the main driving force for positive selection is thought to be heterozygote advantage against malaria in females (*Bienzle et al., 1972*; *Luzzatto, 2015*). These and other recent data confirm that female heterozygotes do have an advantage against cerebral malaria, but raise the question of whether this is a sufficient explanation for current levels of G6PD deficiency, given that female homozygotes and male hemizygotes have increased risk of severe malarial anaemia.

To examine this question, we adapted the theoretical model introduced by Levene (*Levene, 1953*) to model these different selective forces on the frequency of the G6PD+202T allele (see Materials and methods). This necessitated a number of simplifying assumptions. First, we assume that the fitness of this allele is determined only by the effects on severe *P. falciparum* malaria described in *Table 7*, i.e. we do not consider the possible effect of this allele on the risk of other diseases, including those caused by other *Plasmodium* species and haemolytic anaemia in general.

**Table 5.** Calculation of the G6PD deficiency (G6PDd) score. The G6PDd score is calculated on the basis of the haplotypes at the 15 WHO-classified SNPs. For each haplotype, the loss of normal G6PD function is determined according to the WHO severity class (0, I, II, III or IV) of the most severe mutation carried. See Materials and methods for more details. The G6PDd score for each individual is then the average loss of function across the two haplotypes, where males are treated as homozygous females. For example, the assigned loss of normal G6PD function for a haplotype where the most severe mutation is the class III mutation G6PD+202T is 65, so an individual with this haplotype and another with no deficiency mutations will average a G6PDd score of 32.5. Individuals were further categorized on the basis of their G6PDd score as normal G6PDd (G6PDd score = 0); Low G6PDd (0 < G6PDd score < 25, blue boxes); Medium G6PDd (25 ≤ G6PDd score < 50, yellow boxes); High G6PDd (50 ≤ G6PDd score < 85, orange boxes); or Very High (G6PDd ≥ 85, red box). Scores not observed in our study are shown in grey boxes.

| | | | Haplotype 1 | | | | |
| --- | --- | --- | --- | --- | --- | --- | --- |
| | | | Severity class (loss of normal G6PD function) | | | | |
| G6PDd score | | | 0 (0) | IV (20) | III (65) | II (95) | I (100) |
| Haplotype 2 | Severity class (Loss of normal G6PD function) | 0 (0) | 0 | 10 | 32.5 | 47.5 | 50 |
| | | IV (20) | 10 | 20 | 42.5 | 57.5 | 60 |
| | | III (65) | 32.5 | 42.5 | 65 | 80 | 82.5 |
| | | II (95) | 47.5 | 57.5 | 80 | 95 | 97.5 |
| | | I (100 | 50 | 60 | 82.5 | 97.7 | 100 |

Clarke *et al*. eLife 2017;6:e15085. DOI: 10.7554/eLife.15085

Second, the model requires us to specify the rates of mortality due to severe malaria over the past few thousand years. Childhood malaria mortality rates of approximately 5% have been observed in Africa in the recent past (*Greenwood et al., 1987*), but are likely to have been much higher before the advent of malaria control and modern medicine. Cerebral malaria currently has a much higher fatality rate than severe malarial anaemia (in this study, the average case fatality rates were 18% and 5%, respectively), but mortality from severe malarial anaemia is likely to havebeen considerably higher in the past, before the introduction of blood transfusion. For purposes of illustration, we assume that childhood mortality due to malaria has historically been in the range of 10–20%, that all of these deaths were due to cerebral malaria or severe malarial anaemia, and that the mortality rate of cerebral malaria has been approximately twice that of severe malarial anaemia.

**Table 6.** G6PD deficiency score categorical model association test results. Counts, odds ratios (ORs), 95% confidence intervals (95% CI) and *P*-values are presented for the effect of each category of G6PD deficiency (G6PDd) score, compared to individuals with G6PDd score = 0, for all severe malaria cases and for the two subtypes cerebral malaria and severe malarial anaemia. *P*-value shown in bold is for overall model fit. P-value for the overall model fit. Results are shown for all individuals combined at each site and across all sites. Results are adjusted for the sickle-cell trait, ethnicity and sex. n.c., not calculated.

| Sample | G6PDd score category | Controls N (%) | Severe malaria N (%) | OR (95% CI) | P | Cerebral malaria N (%) | OR (95% CI) | P | Severe malarial anaemia N (%) | OR (95% CI) | P |
|---|---|---|---|---|---|---|---|---|---|---|---|
| Females | 0 | 3,761 (46.4) | 2,299 (42.6) | – | – | 672 (42.3) | — | --- | 400 (39.8) | -- | --- |
| All | (0–25) | 2,132 (26.3) | 1,626 (30.1) | 0.98 (0.9–1.08) | 0.72 | 495 (31.2) | 1.02 (0.89–1.17) | 0.81 | 306 (30.4) | 1.08 (0.9–1.29) | 0.41 |
| | (25–50) | 2,001 (24.7) | 1,310 (24.3) | 0.88 (0.8–0.97) | 8.59E-03 | 377 (23.7) | 0.83 (0.71–0.96) | 1.04E-02 | 252 (25.1) | 1 (0.83–1.21) | 0.99 |
| All | (50–85) | 210 (2.6) | 156 (2.9) | 1.02 (0.81–1.28) | 0.85 | 44 (2.8) | 0.93 (0.66–1.31) | 0.67 | 45 (4.5) | 1.71 (1.16–2.53) | 6.86E-03 |
| | ≥85 | 2 (0) | 8 (0.1) | 7.3 (1.43–37.24) | 0.02 | 0 (0) | n.c | n.c. | 2 (0.2) | 9.72 (1.01–93.12) | 0.05 |
| | Overall | 8,106 | 5,399 | | **3.68E-03** | 1,588 | – | **0.06** | 1,005 | – | **0.03** |
| Males | 0 | 5,960 (67.9) | 4,216 (65.1) | — | — | 1,212 (68.4) | – | – | 707 (60) | – | – |
| All | (0–25) | 1,572 (17.9) | 1,303 (20.1) | 1.08 (0.99–1.19) | 0.08 | 342 (19.3) | 1.01 (0.88–1.16) | 0.86 | 227 (19.3) | 1.05 (0.87–1.25) | 0.63 |
| | (25–50) | 0 (0) | 0 (0) | n.c | n.c. | 0 (0) | n.c | n.c. | 0 (0) | n.c | n.c. |
| All | (50–85) | 1,196 (13.6) | 923 (14.3) | 1.05 (0.95–1.16) | 0.35 | 214 (12.1) | 0.79 (0.67–0.94) | 6.31E-03 | 236 (20) | 1.44 (1.2–1.74) | 1.17E-04 |
| | ≥85 | 55 (0.6) | 30 (0.5) | 0.65 (0.4–1.04) | 0.07 | 3 (0.2) | 0.26 (0.08–0.85) | 0.03 | 9 (0.8) | 1.24 (0.58–2.67) | 0.58 |
| | Overall | 8,783 | 6,472 | — | **0.08** | 1,771 | — | **1.49E-03** | 1,179 | – | **1.97E-03** |
| All | 0 | 9,721 (57.6) | 6,515 (54.9) | – | – | 1,884 (56.1) | – | – | 1,107 (50.7) | – | – |
| All | (0–25) | 3,704 (21.9) | 2,929 (24.7) | 1.05 (0.99–1.12) | 0.12 | 837 (24.9) | 1.03 (0.93–1.13) | 0.56 | 533 (24.4) | 1.05 (0.93–1.19) | 0.45 |
| | (25–50) | 2,001 (11.8) | 1,310 (11) | 0.94 (0.86–1.03) | 0.16 | 377 (11.2) | 0.86 (0.75–0.99) | 0.04 | 252 (11.5) | 0.95 (0.8–1.14) | 0.6 |
| | (50–85) | 1,406 (8.3) | 1,079 (9.1) | 1.02 (0.93–1.12) | 0.71 | 258 (7.7) | 0.8 (0.69–0.93) | 2.95E-03 | 281 (12.9) | 1.49 (1.26–1.76) | 2.22E-06 |
| | ≥85 | 57 (0.3) | 38 (0.3) | 0.87 (0.55–1.36) | 0.54 | 3 (0.1) | 0.28 (0.09–0.86) | 0.03 | 11 (0.5) | 1.44 (0.66–3.17) | 0.36 |
| All | Overall | 16,889 | 11,871 | – | **0.14** | 3,359 | – | **2.29E-04** | 2184 | – | **1.02E-04** |

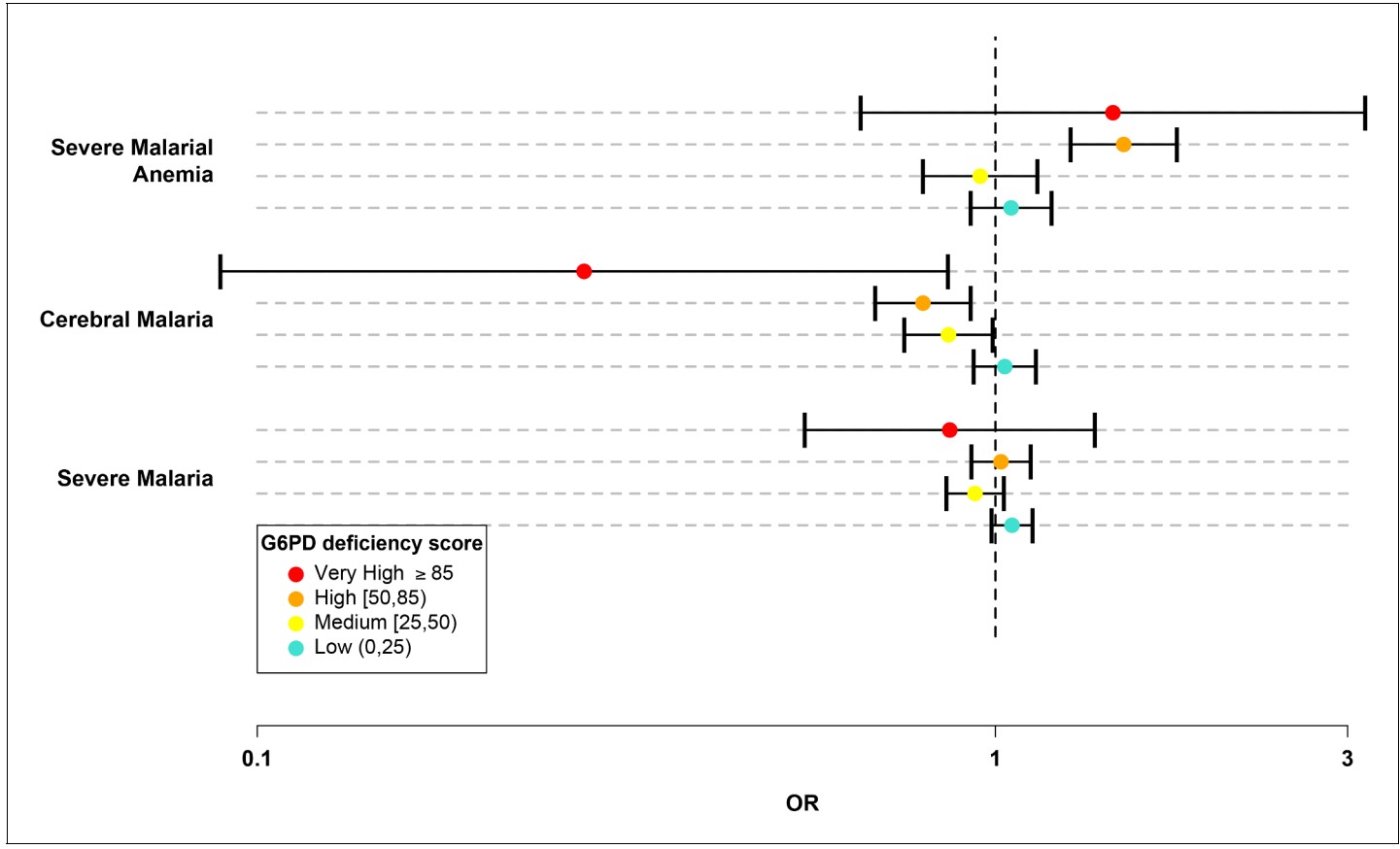

**Figure 2.** G6PD  deficiency score forest plot. Forest plots showing 95% confidence intervals for the effect of each category of G6PDd score on all severe malaria and on two sub-phenotypes, cerebral malaria and severe malarial anaemia. Results are shown for all individuals across all sites combined and are adjusted for the sickle-cell trait, ethnicity and sex. See *Table 6* for details and *Supplementary files 1H–J* for results at all sites for males, females and all individuals combined.

The following source data is available for figure 2:

**Source data 1.** G6PDd score association test results.

We used this simplistic model to investigate the expected change in G6PD+202T allele frequency over time, from a starting value of 0.01, using different assumptions about the proportion of children suffering from severe malaria and about the ratio of severe malaria cases due to cerebral malaria compared to those due to severe malaria anaemia. As shown in *Figure 3*, as long as the proportion of cerebral malaria cases remains greater than approximately 27%, the frequency of G6PD+202T increases over time to an equilibrium value, otherwise it decreases steadily and will be eliminated (dotted black line on red background). We note that this result is not dependent upon the assumed proportion of children suffering from severe malaria; changing this will simply change the time taken to reach equilibrium for any selected ratio of cerebral malaria to severe malarial anaemia cases, not the equilibrium value itself.

Another way of addressing this question is to consider the number of severe malaria deaths that can be attributed to, or prevented by, the presence of G6PD+202T allele. At increasing levels of allele frequency, we would expect the number of cerebral malaria cases to be reduced and the number of severe malarial anaemia cases to be increased, as shown in *Figure 4*. For this analysis, we used the odds ratios observed in this study (*Table 7*) and assumed a baseline risk of death due to cerebral malaria of 200 per 10,000 individuals and baseline risks of death due to severe malarial anaemia of ¼, ½ and 1/3 that of cerebral malaria. When the baseline risk of death due to severe malarial anaemia is half that of cerebral malaria, the equilibrium frequency of G6PD+202T, where

the number of lives saved equals the number of lives lost, is approximately 28%; whereas at lower allele frequencies, there are more lives saved than lost, and at higher allele frequencies, there are more lives lost than saved (solid red and solid blue lines *Figure 4*). As the baseline risk of death due to severe malarial anaemia increases relative to that of cerebral malaria, the equilibrium frequency decreases (solid red and dotted blue lines); and conversely, as the risk of death due to severe malarial anaemia decreases relative to that of cerebral malaria, it increases (solid red and dashed blue lines). These equilibrium values are consistent with G6PD+202T allele frequencies observed in the African populations studied here, which ranged from 11% in Cameroon to 28% in Nigeria (excluding The Gambia where there are other deficiency alleles).

## Discussion

In contrast to other malaria resistance loci such as sickle cell trait and blood group O, which show highly consistent and statistically significant effects across different locations in this large multi-centre study (*Malaria Genomic Epidemiology Network, 2014*), the observed associations with G6PD deficiency are at lower levels of statistical significance and vary between locations (*Supplementary files 1D–J*) (*Manjurano et al., 2015*; *Uyoga et al., 2015*; *Shah et al., 2016*). This is partly because of the genetic complexity of G6PD deficiency, which is affected by multiple allelic variants and has different effects in males and females. Here, we have simplified the problem by estimating each individual's level of enzyme activity from their genotype, thus enabling an aggregated analysis to be performed across both sexes and all allelic variants. Another complicating factor is that severe malaria comprises a variety of different clinical syndromes, of which cerebral malaria and severe malarial anaemia are the most commonly diagnosed causes of fatal outcome in African children. We find significant evidence that increasing levels of G6PD deficiency are associated with decreasing risk of cerebral malaria, but with increased risk of severe malarial anaemia. The majority of evidence comes from the large number of individuals carrying the G6PD+202T allele, but it is supported by evidence from a smaller number of individuals carrying alleles with stronger phenotypic effects. The net result is that there is no significant association between G6PD deficiency and severe

**Table 7.** Fitness arrays suggested by observed OR estimates of effect at G6PD+202 on cerebral malaria and severe malarial anaemia. $A_1$ represents the wild-type allele and $A_2$ the derived allele. In females, odds ratios (ORs) for association with cerebral malaria in derived homozygotes and heterozygotes compared to wild-type homozygotes are 0.87 and ~1, respectively, suggesting a symmetrical heterozygous advantage. In males, the OR for association with cerebral malaria in derived, compared to wild-type, hemizygotes is 0.82, suggesting a moderate fitness difference between male genotypes. For severe malarial anaemia, in females, the ORs in derived homozygotes and heterozygotes, compared to wild-type heterozygotes, are 1.84 and 1, respectively, suggesting selection against derived homozygotes; in males, the OR in derived hemizygotes, compared to wild-type, is 1.48, suggesting selection against derived male hemizygotes. The case fatality rate is estimated to be 20% ($m$ = 0.2) for cerebral and 10% ($m$ = 0.1) for severe malarial anaemia.

| Genotype | Females | | | Males | |
|---|---|---|---|---|---|
| | $A_1A_1$ | $A_1A_2$ | $A_2A_2$ | $A_1$ | $A_2$ |
| | $w_{11}$ | $w_{12}$ | $w_{22}$ | $w_1$ | $w_2$ |
| Cerebral malaria selection | | | | | |
| Observed OR* | 1 | 0.87 (0.75–1) | 1.02 (0.7–1.48) | 1 | 0.82 (0.69–0.97) |
| Fitness Array† | $1-s_f^{CM}$ | 1 | $1-s_f^{CM}$ | $1-s_m^{CM}$ | 1 |
| Fitness ($w$) | 0.974 | 1 | 0.974 | 0.964 | 1 |
| Severe malarial anaemia selection | | | | | |
| Observed OR* | 1 | 0.97 (0.8–1.17) | 1.84 (1.21–2.79) | 1 | 1.48 (1.22–1.8) |
| Fitness Array‡ | 1 | 1 | $1-s_f^{SMA}$ | 1 | $1-s_m^{SMA}$ |
| Fitness ($w$) | 1 | 1 | 0.954 | 1 | 0.968 |

*Odds ratios are estimated from a genotypic model and are adjusted for sickle-cell trait and ethnicity.

†$s_f^{CM} = m(1-OR) = 0.2 (1 0.87) = 0.26$. $s_m^{CM} = 0.2(1-0.82) = 0.036$.

‡$s_f^{SMA} = m (1-1/OR) = 0.1 (1-1/1.84) = 0.046$; $s_m^{SMA} = m(1-1/OR) = 0.1 (1 1/1.48) = 0.032$.

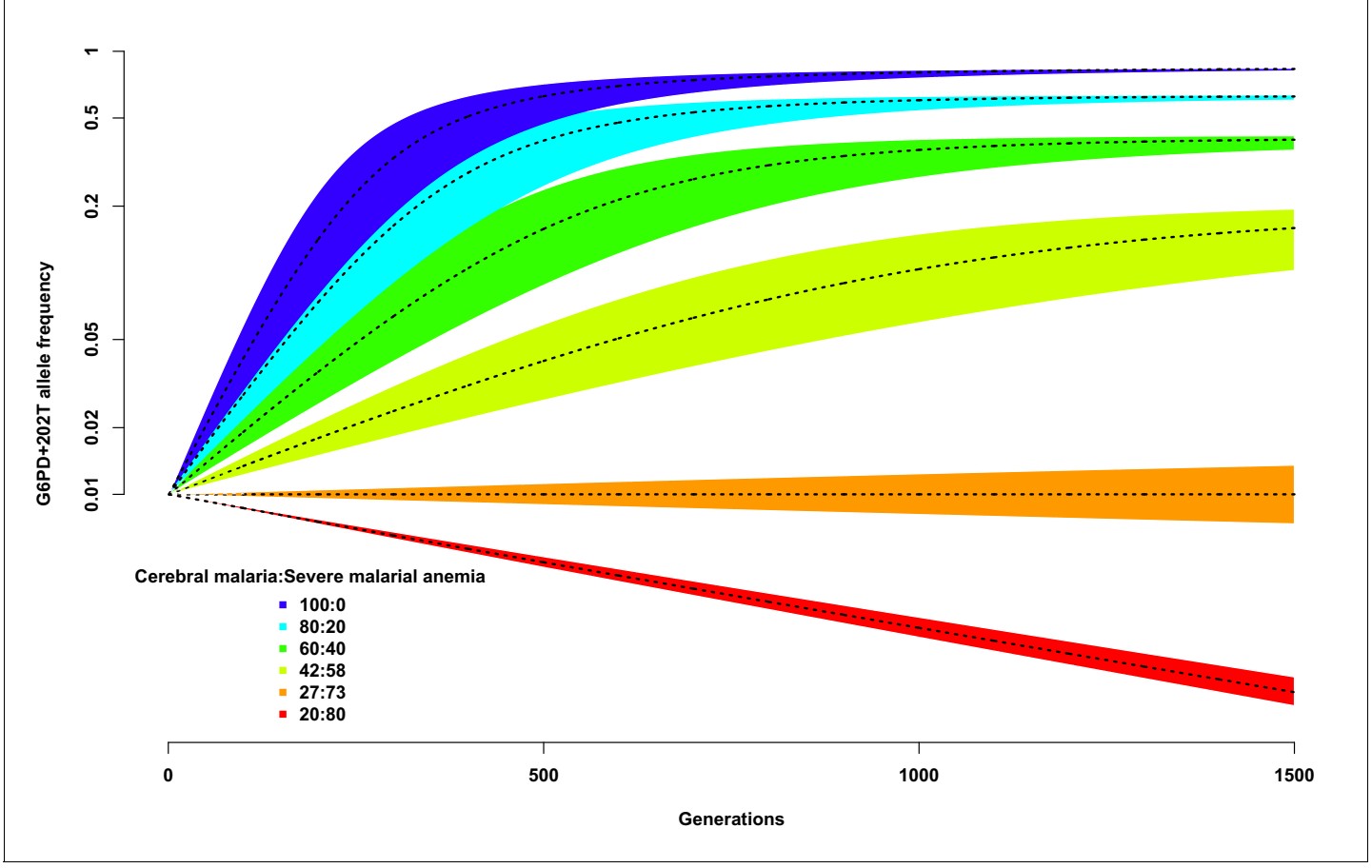

**Figure 3.** Allele frequency of G6PD+202T. Change in the allele frequency of G6PD+202T under the adapted Levene model. We assume an initial frequency of 0.01 in both males and females and that the change in allele frequency depends only on genotype fitness as a consequence of relative exposure to three selective forces: cerebral malaria, severe malarial anaemia and no selection (see Materials and methods). Here we have assumed that, in each generation, 50% of children suffer from severe malaria and show results for different ratios of cerebral malaria to severe malarial anaemia cases, as indicated for each coloured polygon. Within each of these polygons, the solid black lines shows the allele frequencies when thefatality rate of cerebral malaria (severe malarial anaemia) is 20% (10%); the lower bound to 15% (8%); and the upper bound to 25% (12%).

malaria as a whole. We note that the method used here to calculate G6PDd score is based on a relatively crude classificaiton system of known variants (*Yoshida et al., 1971*), and there is a need for additional phenotypic data on the biochemical effects of specific genotypes in order to evaluate the precise relationship between G6PD enzyme function and the risk of different forms of severe malaria.

These findings may help to explain apparent inconsistencies between our findings and those of previous studies, in particular surrounding the question of whether males carrying the G6PD+202T allele are protected against severe malaria. This is the group of individuals in whom the opposing effects of G6PD deficiency on cerebral malaria and severe malarial anaemia are most apparent (*Tables 4* and *6*) and their overall level of protection against severe malaria might therefore depend on local circumstances. However, there remain unexplained differences in the patterns of association that we have observed in different locations (*Supplementary files 1G–I*) and whilst these might be due to the small sample size of local analyses, it is possible that they reflect true biological differences. Severe childhood anaemia in the tropics is often the result of a combination of factors that include malaria, G6PD deficiency, hookworm, micronutrient deficiency, haemoglobinopathies and other infectious diseases (*Calis et al., 2008*; *van Hensbroek et al., 2011*; *Silverberg, 2012*; *Kassebaum et al., 2014*). Further studies are needed to examine the malaria-protective effects of G6PD deficiency under different environmental conditions, and in particular how these affects may

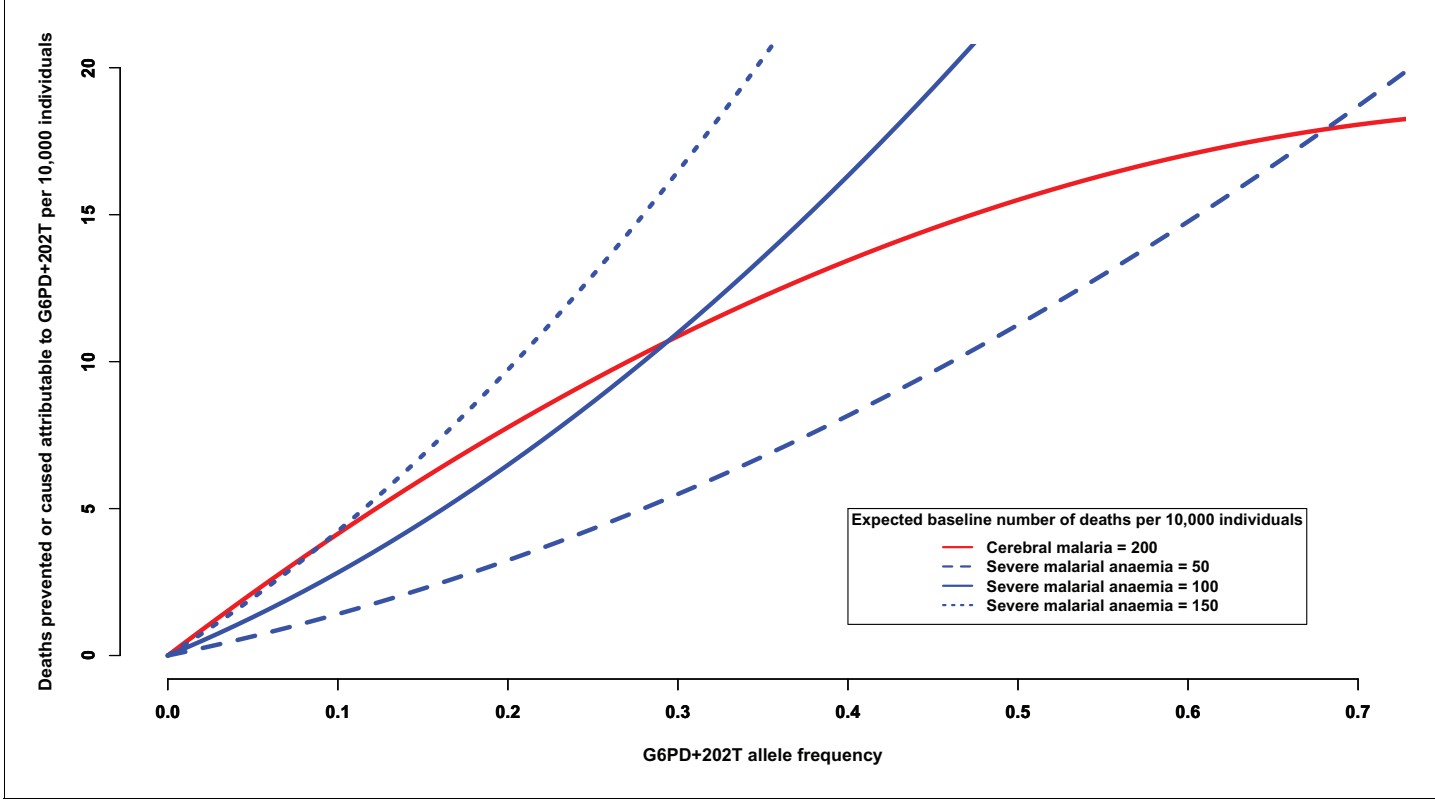

**Figure 4.** Deaths prevented or caused attributable to the genetic effects at G6PD+202. The solid red line shows the number of deaths due to cerebral malaria that are prevented by the presence of G6PD+202T, assuming a baseline risk of death due to cerebral malaria of 200/10,000 individuals. The blue lines show the number of deaths due to severe malarial anaemia that are caused by the presence of G6PD+202T for various baseline risks of death due to severe malarial anaemia, as indicated. The baseline risk of death is the expected number of deaths per 10,000 wild-type individuals, i.e. when the frequency of G6PD+202T is zero. Results are shown as the allele frequency of G6PD+202T increases. Relative risks of disease are estimated by odds ratios observed in this study.

be influenced by other epidemiological variables that act together to cause anaemia in African children.

At first sight, these findings appear to challenge the widely accepted hypothesis that G6PD deficiency has evolved in human populations as a result of balancing selection due to malaria (*Allison, 1960*). They raise the question of how much evolutionary benefit is required to counterbalance the deleterious effects of G6PD deficiency. A small minority of carriers of G6PD deficiency have severe mutations that cause congenital haemolytic anaemia (*Cappellini and Fiorelli, 2008*). The majority are generally asymptomatic apart from an increased risk of neonatal jaundice, unless an event occurs to precipitate an acute episode of haemolytic anaemia (*Beutler, 1994, 2008*; *Valaes, 1994*; *Kaplan and Hammerman, 2010*). Haemolysis-precipitating events include certain drugs, dietary factors (e.g. fava beans) and infectious disease. The antimalarial drug primaquine, which is of great practical importance for the treatment and elimination of *P. vivax* malaria, can precipitate severe and sometimes fatal haemolytic anaemia in G6PD-deficient individuals (*Chu and White, 2016*).

Here we consider a different formulation of the balancing-selection hypothesis, in which the main evolutionary benefit of G6PD deficiency is decreased risk of cerebral malaria, and its main evolutionary disadvantage is increased risk of severe malarial anaemia. We modeled this evolutionary scenario, assuming that childhood mortality resulting from severe malaria has historically been in the range of 10–20%, that all of these deaths were due to cerebral malaria or severe malarial anaemia, and that the mortality rate of cerebral malaria has been approximately twice that of severe malarial anaemia. In this model, G6PD deficiency was positively selected and rose to an equilibrium value as

long as >27% of the severe malaria cases were due to cerebral malaria. We also took an epidemiological approach to this question, by considering the number of severe malaria cases that can be attributed to, or prevented by, the presence of the G6PD+202T allele. If we assume a case fatality rate of 20% for cerebral malaria and 10% for severe malarial anaemia in the absence of treatment then, based on the odds ratios observed in this study, the epidemiological equilibrium state – where the number of lives saved equals the number of lives lost – would be achieved at a G6PD+202T allele frequency of 28%. This is consistent with the 18% median allele frequency that we observe in African populations where G6PD+202T is the only major deficiency allele.

The evolutionary and epidemiological models presented here are simplistic; for example, they do not consider the potential impact of spatial and temporal heterogeneity (*Saunders et al., 2002*) and they make very approximate assumptions about past prevalence and mortality rates. These models ignore other potential costs and benefits of G6PD deficiency, such as mortality due to G6PD-induced haemolytic anaemia that is unrelated to malaria or infant mortality due to G6PD-associated neonatal jaundice, and the possible protective effects of G6PD deficiency against infectious diseases other than *P. falciparum* malaria. Of particular interest for future investigation is the evolutionary significance of the protective effect of G6PD deficiency against *P. vivax* (*Louicharoen et al., 2009*; *Leslie et al., 2010*), a parasite species which appears to have played an important historical role in human evolutionary selection, as evidenced by the selective sweep of the Duffy-negative blood group that confers resistance to *P. vivax* across sub-Saharan Africa (*Livingstone, 1984*). With these caveats, the present findings indicate the need for a new formulation of the balancing-selection hypothesis, in which G6PD polymorphism is maintained in human populations, at least in part, by an evolutionary trade-off between different adverse outcomes of *P. falciparum* infection.

## Materials and methods

### Samples and phenotypes

Cases and controls were recruited from 12 MalariaGEN study sites (*Supplementary file 1A*). Cases of severe malaria were recruited on admission to hospital, usually as part of a larger program of clinical research on malaria, designed and led by local investigators. A population control group, with similar ethnic composition to the cases, was recruited at each of the study sites: cord blood samples were used as population controls at several study sites. The cases and controls included children from 213 different ethnic groups of which 41 comprised at least 5% of individuals at a study site; these included Mandinka, Jola, Wollof, Fula (Gambia); Bambara, Malinke, Peulh, Sarakole (Mali); Mossi (Burkina Faso); Akan, Frarra, Nankana, Kasem (Ghana); Yoruba (Nigeria); Bantu, Semi-Bantu (Cameroon); Chonyi, Giriama, Kauma (Kenya); Mzigua, Wasambaa, Wabondei (Tanzania); Chewa (Malawi); Madang, Sepik (Papua New Guinea); Kinh (Viet Nam). For purposes of analysis, we classified ethnic groups with very small sample size (less than 5% of individuals at any study site) as 'other'. Details of study design at individual sites, collection of clinical and genotype data as well as local epidemiological conditions including malaria endemicity are described in detail elsewhere (*Malaria Genomic Epidemiology Network, 2008*, *2014*) and information about each of the partner studies can be found the Malaria Genomic Epidemiology Network (MalariaGEN) website (see URLs).

The normalised data from each study site were combined to ascertain phenotypes in a standardised manner across the entire data set. A case of severe malaria was defined as an individual admitted to a hospital or clinic with *P. falciparum* parasites in the blood film and with clinical features of severe malaria as defined by WHO criteria (*World Health Organisation, 2000*, *2010*). Severe malaria comprises a number of overlapping syndromes; the most commonly reported being cerebral malaria and severe malarial anaemia. In keeping with standard criteria, cerebral malaria is defined here as a case of severe malaria with Blantyre Coma Score of <3 for a child, or a Glasgow Coma Score of <9 for an adult. Severe malarial anaemia is defined here as a case of severe malaria with a hemoglobin level of <5g/dl or a hematocrit level of <15%. In this report, we do not attempt to classify other severe malaria syndromes such as respiratory distress, which are more complicated to standardise between study sites. A total of 38,926 individual records comprising 16,433 cases of severe malaria and 22,492 controls were obtained from across the 12 study sites (*Supplementary file 1A*). Data were missing for gender in 4% of records and for ethnic group in 2% of records. 33,138 samples were retained for genotyping (*Supplementary file 1B*).

## Ethics

All studies were collected under the approval of the appropriate ethics committees, and all participants gave informed consent. Please refer to *Supplementary file 1A* and the MalariaGEN website (see URLs) for further details.

## Genotyping

As the starting point for analysis of genetic variation at the *G6PD* locus in these samples, we identified 251 known variants in a survey of the literature and polymorphism databases and an additional 57 variants using targeted Sanger sequencing of the *G6PD* locus in 288 individuals comprising 48 randomly selected from Burkina Faso, Cameroon, Kenya, Nigeria, Papua New Guinea and Vietnam. A set of four Sequenom multiplexes was designed for 135 of these SNPs with priority assigned according to proximity to the *G6PD* locus, evidence of a functional role and feasibility of assay design (*Supplementary files 2A,B*). These assays were first tested on a set of 90 HapMap DNA samples each from the CEU and YRI populations. Failing assays were removed and a new set of multiplexes (W1186-W1189) was re-configured containing 107 assays. They were used to genotype a set 100 individuals randomly selected from each of the populations in the study plus the 288 sequenced samples (*Supplementary file 2C*). This provided information on assay performance and population allele frequencies and was used to reduce the number of SNPs carried forward into the main study for reasons of economy and scale: 59 SNPs were selected for a final set of three multiplexes based on location in the exons of *G6PD*, the introns of *G6PD*, the first 5 kb of the 5' sequence of *G6PD*, and the frequency in the screen populations. Nine additional SNPs were subsequently added that were shown to be in long-range-haplotypes with *G6PD* (*Sabeti et al., 2002*) (*Supplementary file 2A*). The reduction from four multiplexes into three multiplexes using existing working assay designs was managed using the Sequenom AssayDesigner software and replexing feature. The final SNPs set comprised 3 multiplexes containing 68 SNPs, of which 54 are assigned to *G6PD* (*Supplementary file 2D*). The two *G6PD* SNPs (rs1050828 and rs1050829) typed in our previous study (*Malaria Genomic Epidemiology Network, 2014*) were also included, resulting in 70 SNPs available for analysis. They ranged from rs763737 at 153,278,307 bp to rs60030796 at 153,836,171 bp (where coordinates are referenced to GRCh37, dbSNP137 and Ensembl build 84). All retained samples were genotyped at these 70 SNPS. Genotype data for these samples were also available within a set of SNPs with prior evidence of association with severe malaria phenotypes, including the sickle hemoglobin (*HbS*) variant, rs334. The above SNPs were genotyped using whole-genome amplified DNA on the Sequenom iPLEX Mass-Array platform (Agena Biosciences, Hamburg, Germany) as described elsewhere (*Malaria Genomic Epidemiology Network, 2014*)

Following genotyping of the final set of 70 SNPs, five SNPs were removed because of missing data in more than 30% of samples. 2,918 individuals were removed because of missing data in more than 20% of the remaining 65 SNPs (*Supplementary file 1B*) and 1,460 individuals were removed because of missing severe malaria classification, leaving 11,871 severe malaria cases and 16,889 controls (*Table 1*), and 65 SNPs (*Supplementary file 1C*), for further analysis.

## Phasing

The X chromosome is paired (diploid) in females and haploid in males. Male haplotypes are therefore phase-known but female haplotypes require phasing. We phased female haplotypes using Shapeit2 (*Delaneau et al., 2012*). In order to verify the accuracy of phasing, half of the male phase-known haplotypes (8,030 of 16,060) were randomly paired to produce 4,015 phase-known ('pseudo-female') samples, and phasing was repeated for a simulated dataset comprising the remaining 8,030 males, all the females and the 4,015 phase-known samples. The resulting phased haplotypes in the 4,015 pseudo-female samples were compared to the phase-known haplotypes.

The phased and phase-known haplotypes from the pseudo female samples were then compared by recording the mismatches at heterozygous loci. For each sample, each phased haplotype was assumed to represent phasing for the phase-known haplotype with the fewest mismatches. In our data, 71% of the haplotypes had less than 10% of the haplotype incorrectly phased across the 65 SNPS. Across the 15 SNPs with a WHO classified G6PD deficiency classification, 83% of haplotypes had less than 10% of the haplotype incorrectly phased. Phasing of the original data is expected to have greater accuracy than the phasing of the simulated data because the simulated data effectively

have 8,030 fewer phase-known samples and 4,105 more phase-unknown samples than the original data.

## G6PD deficiency score

The WHO classification system divides G6PD deficiency variants into four classes of severity (*Yoshida et al., 1971*):

    I. Severe mutations; total loss of normal G6PD function
    II. Intermediate; >90% loss of normal G6PD function
    III. Mild; 40–90% loss of normal G6PD function
    IV. Asymptomatic; 0–40% of normal G6PD function

Using phased data at the 15 SNPs with a WHO classification, a quantitative measure of G6PD deficiency for each individual was calculated. Each phased haplotype of 15 SNPs was assigned to one of the four classes of deficiency according to the most severe deficiency variant carried. The loss of normal G6PD function for the haplotype was then assigned according to that class. The WHO classification specifies a range of values for each class of deficiency and here we used the median of the range:

- Haplotypes carrying no mutation = 0.
- Haplotypes carrying a class I mutation = 100
- Haplotypes carrying a class II mutation = 95
- Haplotypes carrying a class III mutation = 65
- Haplotypes carrying a class IV mutation = 20

The G6PDd score for each *individual* was then calculated as the average of the assigned loss of function over each of their two *haplotypes*, where males are treated as homozygous females (*Table 5*). Observed G6PDd score ranged from 0 to 95. For example, individuals heterozygous for WHO class III variant G6PD+202T and no other mutations scored 65; female homozygotes and male hemizygotes for G6PD+202T scored 80.

Individuals were further categorized on the basis of their G6PDd score as Normal (G6PDd score = 0); Low (0 < G6PDd score <25, blue boxes in *Table 5*); Medium (25 $\leq$ G6PDd score <50, yellow boxes in *Table 5*); High (50 $\leq$ G6PDd score <85, orange boxes in *Table 5*); or Very high (G6PDd score $\geq$85, red boxes in *Table 5*).

In our sample of 28,760 individuals included in the final analysis, there were only six individuals (<0.02%) who carried more than one mutation on the same phased chromosome, making them potentially vulnerable to an incorrectly calculated G6PDd score in the event of incorrect phasing. These six individuals comprised three females from Gambia, who carried two class II mutations, and two females from Ghana (Kumasi) and one from Burkina Faso, who carried two class III mutations.

## Statistical association analysis

We drew on both classical and Bayesian statistical approaches to assess evidence for association. All statistical analyses were primarily written and performed using the statistical software environment R (see URLs).

### Standard fixed effects analyses

In primary analyses, standard fixed effects logistic regression methods were used for tests of association with severe malaria and sub-types at each SNP under additive, dominant, recessive and heterozygous models. Analyses were run separately at each site and then at all sites combined, both for males and females separately and in combination. When examining all sites combined, sites where a SNP was monomorphic were excluded from the analysis. Results were adjusted for sickle hemoglobin (*HbS*), gender and ethnicity. Ethnicity was not shared across any sites. Here, we report on analyses looking for evidence of association with severe malaria and with the two sub-types of fatal severe malaria that are most commonly diagnosed in African children, namely cerebral malaria and severe malarial anaemia. For reasons of sample size, we did not conduct a detailed analysis of other sub-types of severe malaria, or of those individuals who had *both* cerebral malaria and severe malarial anaemia, but summary statistics are available in *Figure 1—source data 1* and *2*. Males were treated as homozygous females and so for analysis of males only, dominant, recessive and additive models are equivalent and the heterozygous model is redundant; in this case, we refer to the model

as hemizygote and note that the odds ratios (OR) correspond to the change in odds of disease for males hemizygous for the derived allele compared to males hemizygous for the ancestral allele. For combined analyses of males and females at X-chromosome SNPS, robust estimates of variance were used to account for the unequal variance (*Clayton, 2008*) and all models are then appropriate. Significance, ORs and 95% confidence intervals were derived from Wald tests applied to regression coefficients. Results are presented with respect to the association between the derived (non-ancestral allele) and the severe malaria phenotype in question. A *P*-value for heterogeneity of the effect of each SNP between severe malaria sub-phenotypes cerebral malaria and severe malarial anaemia was calculated in a logistic regression framework by comparing cerebral-malaria-only cases against severe-malarial-anaemia-only cases. Standard fixed effects logistic regression methods were used to test associations with the G6PDd score with and without adjustment for an additive effect at G6PD +202. To calculate the G6PDd score, the data was phased using SHAPEIT2 (*Delaneau et al., 2013*). See *Phasing* for more details.

## Adjustment for additional risk factors

To investigate the impact of age on risk of severe malaria, we used logistic regression to estimate odds ratios for the risk of cerebral malaria and severe malarial anaemia in cases of various age strata compared to controls. *Figure 1—figure supplement 2* shows forest plots for the effects of G6PD +202T and the G6PDd score on cerebral malaria and severe malarial anaemia in cases in yearly intervals from age 0–6 years and greater than six years, compared to controls. We note that our study is based on a multi-centre collection of severe malaria cases and population controls and was not designed to enable a rigorous study of the effects of age as a possible factor. For this analysis, we have grouped all controls together, assuming that the allele frequency observed in controls is representative of the population allele frequency and that it does not change with age.

To investigate the influence of known genetic factors on association with severe malaria at the *G6PD* gene, we re-analyzed the association between G6PD+202 and the G6PDd score with adjustment for *ABO* (*rs8176719*), *ATP2B4* (*rs10900585*), *FREM3/GYPE* (*rs149914432*) and *HbC* (*rs33930165*). Genotype data for these SNPs for individuals in our study were available from our previous study (*Malaria Genomic Epidemiology Network, 2014*). *Figure 1—figure supplement 3* shows ORs and 95% confidence intervals for each test of association (only shown for tests with *P*-value < 0.05) with and without adjustment for these additional factors.

## Genetic heterogeneity

To allow for possible differences in genetic effects on the two severe malaria sub-phenotypes — cerebral malaria and severe malarial anaemia — within and across sites, we compared different models of association in a Bayesian statistical framework as described in detail elsewhere (*Malaria Genomic Epidemiology Network, 2014*). In brief, we calculated Approximate Bayes Factors (ABF) for nine models of association, allowing for all combinations of fixed and independent effects on phenotypes within and across sites. For each SNP, we assumed that the log odds ratio of association was normally distributed with a mean of zero and standard deviation of 0.4. To model fixed, independent or correlated effects both within and across sites, we set correlation parameters between sub-phenotypes to 1, 0.1 and 0.96, respectively. Multinomial regression was used to make independent maximum likelihood estimates of the effect of each SNP on cerebral malaria and severe malarial anaemia, using cases with either cerebral malaria only or with severe malarial anaemia only. Estimates were adjusted for sex, ethnicity and sickle-cell trait (rs334) and were made separately for males and females, with males being treated as homozygous females; meta-analysis was used to make combined estimates for males and females. The posterior probability of each model was then calculated as the relative size of its ABF compared to that of all other models considered and evidence for heterogeneity was assessed by comparing the posterior probabilities of the selected models.

## Theoretical model for a balanced polymorphism

In order to examine how the different effects observed in males and females in cerebral malaria and severe malarial anaemia could lead to a balanced polymorphism at G6PD+202, we have adapted the theoretical model introduced by Levene (*Levene, 1953*) to model the allele frequency over time

of an X-linked locus allowing for multiple selective forces. In this simple model, it is assumed that the population mates at random and that the progeny then settle at random into one of multiple mutually exclusive 'niches'. There is then differential mortality in each niche before random mating across all niches produces the next generation. Here, we are adapting Levene's concept of a 'niche' to represent a proportion of the population suffering from a particular severe malaria sub-phenotype, and then modelling the allele frequency as a combination of the selective effects from multiple niches over time and space.

We now derive the conditions for a polymorphism as a result of a different selection regime in each niche at an X-linked locus. Consider an X-linked biallelic locus with alleles $a$ and $A$. For the $i^{th}$ niche: let $p_{m,i}$ and $p_{f,i}$ be the frequencies of $a$ in males and females; $q_{m,i}$ and $q_{f,i}$ the corresponding frequencies for $A$; $w_{aa.i}$, $w_{aA.i}$ and $w_{AA.i}$ the fitness of females carrying $aa$, $aA$ and $AA$ genotypes, respectively; and $w_a$ and $w_A$ the fitness of males carrying $a$ and $A$ alleles. Recursion equations give the next generation's male and female allele frequencies as a function of the current generation's (*Haldane and Jayakar, 1964*):

$$q'_{m,i} = \frac{q_f w_{A,i}}{p_f w_{a,i} + q_f w_{A,i}} \tag{1}$$

$$q'_{f,i} = \frac{1}{2} \cdot \frac{(p_f q_m + q_f p_m) w_{aA,i} + 2 q_f q_m w_{AA,i}}{p_f p_m w_{aa,i} + (p_f q_m + q_f p_m) w_{aA,i} + q_f q_m w_{AA,i}} \tag{2}$$

Suppose in general that there are $n$ niches, and that $c_i$ is the proportion of individuals from niche $i$, then the mean frequency of the next generation's male and female alleles is

$$q'_m = \sum_{i=1}^{n} c_i q'_{m,i} \tag{3}$$

$$q'_f = \sum_{i=1}^{n} c_i q'_{f,i} \tag{4}$$

If there are equal numbers of males and females in the population, then two-thirds of the alleles are in the females and one-third are in males for this X-linked locus and the mean allele frequency in the next generation will be

$$q' = \frac{2}{3} q'_f + \frac{1}{3} q'_m \tag{5}$$

When allele $A$ is rare, $q_m$ and $q_f$ are close to zero, and *Equation 3* and *Equation 4* approximate to

$$q'_m = q_f \sum_{i=1}^{n} \frac{c_i w_{A,i}}{w_{a,i}} \tag{6}$$

$$q'_f = \frac{(q_m + q_f)}{2} \sum_{i=1}^{n} \frac{c_i w_{aA,i}}{w_{aa,i}} \tag{7}$$

The $A$ allele can invade when the larger eigenvalue of the Jacobian of this system of equations is greater than one (*Hartl and Clark, 1989*; *Patten and Haig, 2009*): the approximate invasion condition for allele $A$ is:

$$\left(1 + \sum_{i=1}^{n} c_i \frac{w_{a,i}}{w_{A,i}}\right) \left(\sum_{i=1}^{n} c_i \frac{w_{aA,i}}{2 w_{AA,i}}\right) > 1 \tag{8}$$

Similarly, it can be shown that the approximate invasion condition for allele $a$ is:

$$\left(1 + \sum_{i=1}^{n} c_i \frac{w_{A,i}}{w_{a,i}}\right) \left(\sum_{i=1}^{n} c_i \frac{w_{aA,i}}{2 w_{aa,i}}\right) > 1 \tag{9}$$

A balanced polymorphism is possible for an X-linked allele under spatial variation when *Equation 8* and *Equation 9* are simultaneously satisfied.

In our application, we have three 'niches', or equivalently, a combination of three selective forces: cerebral malaria, severe malarial anaemia and no malaria. Essentially, the proportion of individuals in each niche is a proxy for the proportion of children suffering from the corresponding severe malaria subphenotype, or not suffering from severe malaria. In the absence of severe malaria, we assume that all selection coefficients are zero, otherwise, we estimate the selection coefficients for the different genotypes from our estimated ORs (*Table 7*). If the OR indicates a protective effect (OR <1), then the estimated selection coefficient against disease for that genotype can be estimated as s = m (1–OR), where *m* is the case fatality rate (*Hill, 1991*; *Hedrick, 2004*). Hedrick (*Hedrick, 2011*) suggests a value of *m* = 0.1 for severe malaria to take into account mortality over many generations when malaria was endemic as well as the cumulative mortality of individuals with malaria prior to reproduction. As reports estimate that cerebral malaria has a mortality rate that is twice that of severe malarial anaemia (*Thuma et al., 2011*), we use *m* = 0.1 for severe malarial anaemia and *m* = 0.2 for cerebral malaria. In African studies of children aged 0–5 years (*Severe malaria, 2014*), it has been estimated that the current prevalence of coma and anaemia in severe *P. falciparum* malaria is 38% and 52%, respectively, indicating a ratio of 42:58 of cerebral malaria to severe malarial anaemia cases. The change in the frequency of G6PD+202T over time for this ratio and others, assuming an initial frequency of 0.01 and that 50% of children suffer from severe malaria, is shown in *Figure 3*.

## Deaths prevented or caused attributable to genetic effects at G6PD +202

The risk ratio (RR), which can be estimated by the odds ratio, is a measure of the strength of association between one or more risk factors (here the genotypes) and an outcome (here severe malaria). In order to provide information about the importance of the risk factors at the population level, the frequency of the risk factors must be taken into consideration. Let D be an indicator of disease (D = 1 if affected and D = 0 otherwise). The total risk of disease is

$$P(D=1)=f\sum_{g}RR_{g}P(g) \tag{10}$$

where *g* denotes the set of risk factors, *f* the baseline risk of disease and $RR_g$ the relative risk of risk factor *g* compared to the baseline.

Consider a genotype *g* at an X-linked disease susceptibility locus with alleles *a* and *A*, where allele A is considered to be the risk allele. Let $p_A$ denote the frequency of allele A, assumed to be the same in both males and females, and $f_g$= Pr(D = 1|g) the risk of disease in individuals carrying genotype *g*. Furthermore, let RR$_g$ = $f_g/f_{aa}$ denote the relative risk in females carrying genotypes *g* = *aA*, *AA*, respectively, compared to those carrying the baseline genotype *aa* and, similarly, let RR$_A$ = $f_A/f_a$ denote the relative risk in males carrying genotype *A* compared to those carrying the baseline genotype *a*. Assuming Hardy-Weinberg equilibrium and applying (*Equation 10*) above

$$P(D=1|female) = f_{aa}\left((1-p_A)^2+ 2RR_{aA}p_A(1-p_A) + RR_{AA}p_A^2\right) \tag{11}$$

$$P(D=1|male) = f_a((1-p_A) + RR_Ap_A) \tag{12}$$

Using (*Equations 11 and 12*) and assuming equal numbers of males and females, a common baseline risk of disease, f = $f_{aa}$ = $f_a$, and common case fatality rate (*m*) in both males and females, it is then straightforward to show that the expected number of deaths ($N_D$) due to disease in a population of size N is

$$N_D = \frac{1}{2}\left((1-p_A)(2-p_A) + 2RR_{aA}p_A(1-p_A) + RR_{AA}p_A^2 + RR_Ap_A\right) \times N \times f \times m \tag{13}$$

Note that *Nfm* is the expected number of deaths due to disease in a population of size *N* when the frequency of the risk allele A is zero ($p_A$ = 0) and is referred to as the baseline number of deaths per population size N, or baseline risk. Number of deaths for a given frequency that are attributable to allele *A* is then calculated as $N_D$ – Nfm. When allele *A* is protective, $N_D$ – Nfm <0, indicating deaths prevented that are attributable to allele *A*; when allele *A* is risk, $N_D$ – Nfm > 0, indicating deaths caused that are attributable to allele *A*.

Assuming RRs of 1.84 (1.48) for risk of severe malarial anaemia in females homozygous (males hemizygous) for G6PD+202T compared to wild-type homozygotes (hemizygotes) (per OR estimates

in *Table 7*) and null risks otherwise, we calculate $N_D$ for various values of $Nfm$ as the allele frequency $p_A$ varies. We also calculate similar values for cerebral malaria assuming RRs of 0.87 (0.82) for risk of cerebral malaria in females heterozygous (males hemizygous) for G6PD+202T compared to wild-type homozygotes (hemizygotes) (per OR estimates in *Table 7*) and null risks otherwise. *Figure 4* shows $|N_D - Nfm|$ as a function of $p_A$ for selected baseline risks.

## Acknowledgements

The MalariaGEN Project is supported by the Wellcome Trust (077383/Z/05/Z) and the Bill and Melinda Gates Foundation through the Foundation for the National Institutes of Health (566) as part of the Grand Challenges in Global Health Initiative. The Resource Centre for Genomic Epidemiology of Malaria is supported by the Wellcome Trust (090770/Z/09/Z). This research was supported by the UK Medical Research Council (G0600718; G0600230), the Wellcome Trust Biomedical Ethics Enhancement Award (087285) and Strategic Award (096527). DK receives support from the UK Medical Research Council (G19/9). CCAS was supported by a Wellcome Trust Career Development Fellowship (097364/Z/11/Z). The Wellcome Trust also provides core awards to The Wellcome Trust Centre for Human Genetics (090532/Z/09/Z) and the Wellcome Trust Sanger Institute (077012/Z/05/Z).

Mali. The Mali MRTC – BMP group is supported by an ICDR grant of NIAID-NIH to the University of Maryland and the University of Bamako and by the Mali-NIAID/NIH ICER at USTTB, Mali.

Nigeria. Contributions from Nigeria were supported financially by a grant within the BioMalPar European Network of Excellence (LSHP-CT-2004–503578).

Cameroon. EA received partial funding from the European Community's Seventh Framework Programme (FP7/2007–2013) under grant agreement N° 242095 – EVIMalaR and the Central African Network for Tuberculosis, HIV/AIDS and Malaria (CANTAM) funded by the European and Developing Countries Clinical Trials Partnership (EDCTP).

Kenya. TNW is funded by Senior Fellowships from the Wellcome Trust (076934/Z/05/Z and 091758/Z/10/Z) and through the European Community's Seventh Framework Programme (FP7/2007–2013) under grant agreement N° 242095 – EVIMalaR. The KEMRI-Wellcome Trust Programme is funded through core support from the Wellcome Trust. This paper is published with the permission of the Director of KEMRI. CN is supported through a strategic award to the KEMRI-Wellcome Trust Programme by the Wellcome Trust (084538).

Tanzania. The Joint Malaria Programme (JMP) is a collaboration between the National Institute for Medical Research (NIMR), Kilimanjaro Christian Medical College (KCMC), the London School of Hygiene and Tropical Medicine (LSHTM), and the Centre for Medical Parasitology, University of Copenhagen (CMP). Data collection was funded by the UK Medical Research Council (G9901439) and by the European Commission (Europaid: SANTE/2004/078–607).

Vietnam. We would like to thank all the Vietnamese individuals who agreed to provide samples for this study. We acknowledge the work of the clinical staff from the Hospital of Tropical Diseases, HCMC and Phuoc Long and Dong Xoai District Hospitals in Binh Phuoc province, Viet Nam, who initially diagnosed and studied the patients with severe malaria. We would like to thank Dr. Nguyen Thi Hieu and his staff from Hung Vuong Obstetric Hospital for the collection of the cord blood controls. The clinical component of this study was funded through the Wellcome Trust Major Overseas Program in Vietnam (089276/Z/09/Z).

Papua New Guinea. LM was supported through the Basser (Royal Australasian College of Physicians) and National Health and Medical Research Council (NHMRC) scholarships. ML was supported through a Fogarty Foundation Scholarship. TMED was supported through an NHMRC practitioner fellowship.

## Additional information

### Competing interests

JF: Director of the Wellcome Trust, one of the three founding funders of eLife. The other authors declare that no competing interests exist.

## Funding

| Funder | Grant reference number | Author |
| --- | --- | --- |
| Wellcome Trust | 090770/Z/09/Z | Geraldine M Clarke<br>Kirk Rockett<br>Katja Kivinen<br>Christina Hubbart<br>Anna E Jeffreys<br>Kate Rowlands<br>Síle F Molloy<br>Angeliki Kerasidou<br>Victoria J Cornelius<br>Lee Hart<br>Gavin Band<br>Chris CA Spencer<br>Dominic P Kwiatkowski |
| Foundation for the National Institutes of Health | Bill and Melinda Gates Foundation Grand Challenges in Global Health 566 | Geraldine M Clarke<br>Kirk Rockett<br>Katja Kivinen<br>Christina Hubbart<br>Anna E Jeffreys<br>Kate Rowlands<br>Muminatou Jallow<br>David J Conway<br>Kalifa A Bojang<br>Margaret Pinder<br>Stanley Usen<br>Fatoumatta Sisay-Joof<br>Giorgio Sirugo<br>Ousmane Toure<br>Mahamadou A Thera<br>Salimata Konate<br>Sibiry Sissoko<br>Amadou Niangaly<br>Belco Poudiougou<br>Valentina D Mangano<br>Edith C Bougouma<br>Sodiomon B Sirima<br>David Modiano<br>Lucas N Amenga-Etego<br>Anita Ghansah<br>Kwadwo A Koram<br>Michael D Wilson<br>Anthony Enimil<br>Jennifer Evans<br>Olukemi K Amodu<br>Subulade Olaniyan<br>Tobias Apinjoh<br>Regina Mugri<br>Andre Ndi<br>Carolyne M Ndila<br>Sophie Uyoga<br>Alexander Macharia<br>Norbert Peshu<br>Thomas N Williams<br>Alphaxard Manjurano<br>Nuno Sepúlveda<br>Taane G Clark<br>Eleanor Riley<br>Chris Drakeley<br>Hugh Reyburn<br>Vysaul Nyirongo<br>David Kachala<br>Malcolm Molyneux<br>Sarah J Dunstan<br>Nguyen Hoan Phu<br>Nguyen Ngoc Quyen<br>Cao Quang Thai<br>Tran Tinh Hien<br>Laurens Manning<br>Moses Laman<br>Peter Siba<br>Harin Karunajeewa |

|  |  | Steve Allen<br>Angela Allen<br>Timothy ME Davis<br>Pascal Michon<br>Ivo Mueller<br>Síle F Molloy<br>Susana Campino<br>Angeliki Kerasidou<br>Victoria J Cornelius<br>Lee Hart<br>Shivang S Shah<br>Gavin Band<br>Chris CA Spencer<br>Tsiri Agbenyega<br>Eric Achidi<br>Ogobara K Doumbo<br>Jeremy Farrar<br>Kevin Marsh<br>Terrie Taylor<br>Dominic P Kwiatkowski |
| --- | --- | --- |
| Wellcome Trust | 097364/Z/11/Z (Spencer) | Chris CA Spencer |
| European Commission | LSHP-CT-2004-503578 (Nigeria) | Olukemi K Amodu<br>Subulade Olaniyan |
| National Institute of Allergy and Infectious Diseases |  | Ousmane Toure<br>Mahamadou A Thera<br>Salimata Konate<br>Sibiry Sissoko<br>Amadou Niangaly<br>Belco Poudiougou |
| European Commission | Europaid: SANTE/2004/078-607 | Alphaxard Manjurano<br>Nuno Sepúlveda<br>Taane G Clark<br>Eleanor Riley<br>Chris Drakeley<br>Hugh Reyburn<br>Timothy ME Davis |
| European and Developing Countries Clinical Trials Partnership |  | Eric Achidi |
| National Health and Medical Research Council |  | Laurens Manning<br>Timothy ME Davis |
| Fogarty International Center |  | Moses Laman |
| Seventh Framework Programme | 242095 | Thomas N Williams<br>Eric Achidi |
| Wellcome Trust | 076934/Z/05/Z and 091758/Z/10/Z | Sophie Uyoga<br>Alexander Macharia<br>Thomas N Williams |
| Wellcome Trust | 084538 | Carolyne M Ndila |
| Wellcome Trust | KEMRI Core Support | Carolyne M Ndila<br>Sophie Uyoga<br>Alexander Macharia<br>Norbert Peshu<br>Thomas N Williams<br>Kevin Marsh |
| Wellcome Trust | WT Major Overseas Program in Vietnam 089276/Z/09/Z | David Kachala<br>Nguyen Hoan Phu<br>Nguyen Ngoc Quyen<br>Cao Quang Thai<br>Tran Tinh Hien<br>Jeremy Farrar |
| Royal Australasian College of Physicians |  | Laurens Manning |
| Wellcome Trust | MalariaGEN Project 077383/Z/05/Z | Geraldine M Clarke<br>Kirk Rockett |

Christina Hubbart
Anna E Jeffreys
Kate Rowlands
Muminatou Jallow
David J Conway
Kalifa A Bojang
Margaret Pinder
Stanley Usen
Fatoumatta Sisay-Joof
Giorgio Sirugo
Ousmane Toure
Mahamadou A Thera
Salimata Konate
Sibiry Sissoko
Amadou Niangaly
Belco Poudiougou
Valentina D Mangano
Edith C Bougouma
Sodiomon B Sirima
David Modiano
Lucas N Amenga-Etego
Anita Ghansah
Kwadwo A Koram
Michael D Wilson
Anthony Enimil
Jennifer Evans
Olukemi K Amodu
Subulade Olaniyan
Tobias Apinjoh
Regina Mugri
Andre Ndi
Carolyne M Ndila
Sophie Uyoga
Alexander Macharia
Norbert Peshu
Thomas N Williams
Alphaxard Manjurano
Nuno Sepúlveda
Taane G Clark
Eleanor Riley
Chris Drakeley
Hugh Reyburn
Vysaul Nyirongo
David Kachala
Malcolm Molyneux
Sarah J Dunstan
Nguyen Hoan Phu
Nguyen Ngoc Quyen
Cao Quang Thai
Tran Tinh Hien
Laurens Manning
Moses Laman
Peter Siba
Harin Karunajeewa
Steve Allen
Angela Allen
Timothy ME Davis
Pascal Michon
Ivo Mueller
Síle F Molloy
Susana Campino
Angeliki Kerasidou
Victoria J Cornelius
Lee Hart
Shivang S Shah
Gavin Band
Chris CA Spencer
Tsiri Agbenyega
Eric Achidi
Ogobara K Doumbo
Jeremy Farrar
Kevin Marsh
Terrie Taylor
Dominic P Kwiatkowski

| Wellcome Trust | WT Centre for Human Genetic Core 090532/Z/09/Z | Geraldine M Clarke<br>Kirk Rockett<br>Christina Hubbart<br>Anna E Jeffreys<br>Kate Rowlands<br>Síle F Molloy<br>Angeliki Kerasidou<br>Victoria J Cornelius<br>Lee Hart<br>Shivang S Shah<br>Gavin Band<br>Chris CA Spencer<br>Dominic P Kwiatkowski |
|---|---|---|
| Wellcome Trust | WT Sanger Institute Core 077012/Z/05/Z | Katja Kivinen<br>Susana Campino<br>Dominic P Kwiatkowski |
| Wellcome Trust | 087285;096527 | Angeliki Kerasidou |
| Medical Research Council | Tanzania Joint Collaboration G9901439 | Alphaxard Manjurano<br>Nuno Sepúlveda<br>Taane G Clark<br>Eleanor Riley<br>Chris Drakeley<br>Hugh Reyburn<br>Timothy ME Davis |
| Medical Research Council | G0600718; G0600230 | Kirk Rockett<br>Dominic P Kwiatkowski |
| Medical Research Council | G19/9 | Dominic P Kwiatkowski |

The funders had no role in study design, data collection and interpretation, or the decision to submit the work for publication.

## Author contributions

GMC, Writing group; Analysis; KRoc, Writing group; Project management; Sample clinical data collection and management; Sample processing genotyping and management; Analysis; KK, CH, AEJ, KRow, BP, ECB, Sample processing, genotyping and management; MJ, FS-J, GS, OT, ANi, VDM, LNA-E, AG, AE, SO, RM, ANd, SUy, AMac, AMan, NS, TGC, VN, DK, NHP, NNQ, CQT, LM, ML, SFM, SC, LH, SSS, Sample clinical data collection and management; Sample processing, genotyping and management; DJC, KAB, MP, SBS, KAK, MDW, TNW, ER, CD, MM, TTH, PS, TMED, IM, AK, VJC, EA, OKD, JF, KM, TT, Project management; SUs, JE, CMN, Sample clinical data collection and management; MAT, SK, SS, Project management; sample clinical data collection and management.; DM, OKA, TAp, HK, SA, AA, TAg, Project management; Sample clinical data collection and management; Sample processing, genotyping and management; NP, HR, SJD, PM, Project management; Sample clinical data collection and management; GB, CCAS, Analysis; DPK, Writing group; Project management; Analysis

## Author ORCIDs

Geraldine M Clarke, http://orcid.org/0000-0001-7249-0289
Kirk Rockett, http://orcid.org/0000-0002-6369-9299
Mahamadou A Thera, http://orcid.org/0000-0002-2679-035X
Taane G Clark, http://orcid.org/0000-0001-8985-9265

## Ethics

Human subjects: All studies were collected under the approval of the appropriate ethics committees, and all participants gave informed consent. Please refer to Supplementary file 1A and the Malaria-GEN website (see URLs) for further details. Please refer to http://www.malariagen.net/community/ethics-governance

## Additional files

**Supplementary files**

• Supplementary file 1. (A) Summary of study designs of contributing partner studies to MalariaGEN Consortium Project 1 (CP1). (B) Genotyped sample distribution. (C) Summary of 65 SNPs selected for analysis and successfully genotyped. (D) G6PD+202 female association test results. (E) G6PD+202 male association test results. (F) G6PD+202 all individuals association test results. (G) G6PD-deficiency score female categorical model female association test results. (H) G6PD-deficiency score categorical model male association test results. (I) G6PD-deficiency score categorical model all individuals association test results. (J) G6PD-deficiency score additive model association test results.

• Supplementary file 2. (A) SNP selection across *G6PD* region for genotyping. (B) SpectroDESIGNER assay design file for 135 *G6PD* locus SNPs in four multiplexes. (C) SpectroDESIGNER assay design file for 107 *G6PD* locus SNPs in four multiplexes. (D) SpectroDESIGNER assay design file for 68 G6PD locus SNPs in three multiplexes.

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
