## [Decision Letter]

Thank you for submitting your article "Multi-centre analysis of the association between severe malaria and multiple forms of G6PD deficiency" for consideration by *eLife*. Your article has been reviewed by three peer reviewers, and the evaluation has been overseen by a Reviewing Editor, Urszula Krzych, and Prabhat Jha as the Senior Editor. The reviewers have opted to remain anonymous.

The reviewers have discussed the reviews with one another and the Reviewing Editor has drafted this decision to help you prepare a revised submission.

Summary:

The authors previously published their observations concerning an association between G6PD deficiency alleles and decreased risk of cerebral malaria risk as well as increased risk of severe malarial anaemia. In this paper, the authors examine the effect of other forms, both known and novel, of G6PD deficiently on susceptibility to malaria. This study represents a multi-centre analysis, including 11,871 cases of severe malaria caused by *Plasmodium falciparum* and 16,889 population controls from 11 countries representing Africa, Asia and Oceania. The observations revealed that when considered as combined, all forms of G6PD deficiency associate with decreased risk of cerebral malaria. In contrast, the risk of several malarial anaemia increases, with the loss of G6PD function. This study of the relationship of the different forms of G6PD to malaria is timely as the current results will help to resolve lingering uncertainties of the health benefits and costs of carrying these alleles. Overall this is an interesting study that includes new data and interesting modeling of population effects. The results provide strong evidence that malaria was a driving force in positive selection for G6PD trait, or at least for G6PD+202T. The manuscript is well written with generally clear explanations of the methods employed and results obtained.

Essential revisions:

Although the reviewers expressed some concern about the significance of these findings vis-à-vis the authors' previously published works, the additional explanations provided separately upon editor’s request mitigate these issues. However, the authors need to include some aspects of the explanations in the appropriate sections of the revised version of the manuscript.

In addition, please comment:

1) About the other common genetic resistance factors that might have influenced the observed outcome.

2) Age as a possible factor.

3) A broader influence of G6PD on susceptibility to anemia.

---

## [Author Response]

*[…] Essential revisions:*

*Although the reviewers expressed some concern about the significance of these findings vis-à-vis the authors' previously published works, the additional explanations provided separately upon editor’s request mitigate these issues. However, the authors need to include some aspects of the explanations in the appropriate sections of the revised version of the manuscript.*

The Abstract, Introduction and Discussion have been extensively revised to highlight fundamental questions arising from the published literature, including our own recent work, and to set out more clearly and concisely the purpose of this study and the significance of the findings. For example:

Introduction, final paragraph: “The present study had two main aims. The first aim was to perform a more comprehensive analysis of how an individual’s level of G6PD deficiency affects the risk of severe malaria in general, and of cerebral malaria and severe malarial anaemia in particular. […] The second aim of the study was to explore evolutionary and epidemiological models based on these new findings to re-examine the hypothesis that *Plasmodium falciparum* malaria is a major force for G6PD balancing selection.”

Discussion, first paragraph: “In contrast to other malaria resistance loci such as sickle cell trait and blood group O, which show highly consistent and statistically significant effects across different locations in this large multi-centre study, the observed associations with G6PD deficiency are at lower levels of statistical significance and vary between locations. […] The majority of evidence comes from the large number of individuals carrying the G6PD+202T allele, but it is supported by evidence from a smaller number of individuals carrying alleles with stronger phenotypic effects.”

Discussion, final three paragraphs: “At first sight, these findings appear to challenge the widely-accepted hypothesis that G6PD deficiency has evolved in human populations as a result of balancing selection due to malaria (Allison, 1960). […] With these caveats, the present findings indicate the need for a new formulation of the balancing selection hypothesis, in which G6PD polymorphism is maintained in human populations, at least in part, by an evolutionary tradeoff between different adverse outcomes of *P. falciparum* infection.”

In addition, the presentation of our models for the investigation of whether G6PD polymorphism is maintained by balancing selection due to malaria has been simplified. This has not resulted in any material changes to results or overall conclusions.

*In addition, please comment:*

*1) About the other common genetic resistance factors that might have influenced the observed outcome.*

New analyses of this question are presented in the Results, with a new supplementary figure (Figure 1—figure supplement 3), and in the Materials and methods section “Adjustment for additional risk factors”.

Results, Analysis of association with individual variants, second paragraph: “In these analyses we corrected for the effects of sickle-cell trait because of its strong protective effect against both cerebral malaria and severe malarial anaemia. We also examined the ABO, ATP2B4, FREM3/GYPE loci which have well-validated protective effects against severe and found no evidence that they affected the association of G6PD variants with severe malaria (Figure 1—figure supplement 3).”

*2) Age as a possible factor.*

New analyses of this question are presented in the Results, with a new supplementary figure (Figure 1—figure supplement 2), and in the Materials and methods section “Adjustment for additional risk factors.”

Results, Analysis of association with individual variants, fourth paragraph: “In this dataset, the median age of severe malarial anaemia cases was 1.8 years and that of cerebral malaria was 3.4 years (Table 1). […] We conclude from these findings, and from more detailed estimates of age-specific effects shown in Figure 1—figure supplement 2, that the effects of G6PD on risk of cerebral malaria and severe malarial anaemia are not significantly affected by age.”

*3) A broader influence of G6PD on susceptibility to anemia.*

The discussion has been extensively revised including a broader perspective on the complex relationship between G6PD deficiency and anaemia:

Discussion, second paragraph: “Severe childhood anaemia in the tropics is often the result of a combination of factors that include malaria, G6PD deficiency, hookworm, micronutrient deficiency, haemoglobinopathies and other infectious diseases. Further studies are needed to examine the malaria-protective effects of G6PD deficiency under different enviromental conditions, and in particular how this may be affected by other epidemiological variables that act together to cause anaemia in African children.”

Discussion, third paragraph: “At first sight, these findings appear to challenge the widely-accepted hypothesis that G6PD deficiency has evolved in human populations as a result of balancing selection due to malaria. […] The antimalarial drug primaquine, which is of great practical importance for the treatment and elimination of *P. vivax* malaria, can precipitate severe and sometimes fatal haemolytic anaemia in G6PD deficient individuals.”